# A CpG island-encoded mechanism protects genes from premature transcription termination

Amy L. Hughes[1], Aleksander T. Szczurek[1], Jessica R. Kelley ●[1], Anna Lastuvkova[1], Anne H. Turberfield[1], Emilia Dimitrova ●[1], Neil P. Blackledge[1] & Robert J. Klose ●[1]✉

Transcription must be tightly controlled to regulate gene expression and development. However, our understanding of the molecular mechanisms that influence transcription and how these are coordinated in cells to ensure normal gene expression remains rudimentary. Here, by dissecting the function of the SET1 chromatin-modifying complexes that bind to CpG island-associated gene promoters, we discover that they play a specific and essential role in enabling the expression of low to moderately transcribed genes. Counterintuitively, this effect can occur independently of SET1 complex histone-modifying activity and instead relies on an interaction with the RNA Polymerase II-binding protein WDR82. Unexpectedly, we discover that SET1 complexes enable gene expression by antagonising premature transcription termination by the ZC3H4/WDR82 complex at CpG island-associated genes. In contrast, at extragenic sites of transcription, which typically lack CpG islands and SET1 complex occupancy, we show that the activity of ZC3H4/WDR82 is unopposed. Therefore, we reveal a gene regulatory mechanism whereby CpG islands are bound by a protein complex that specifically protects genic transcripts from premature termination, effectively distinguishing genic from extragenic transcription and enabling normal gene expression.

Precise control of gene expression is essential for cell viability and normal development. At the most basic level, gene expression is controlled by transcription factors that recognise specific DNA sequences in gene regulatory elements and shape how RNA Polymerase II (RNA Pol II) initiates transcription from the core gene promoter[1]. Beyond these DNA sequence-encoded mechanisms, transcription is also influenced by chromatin or epigenetic states at gene promoters and by mechanisms that regulate elongation of RNA Pol II (reviewed in[1–6]). However, we understand far less about how these additional influences on transcription are coordinated with initiation to control gene expression.

CpG islands (CGIs) are associated with the majority of vertebrate gene promoters and are distinguished from the rest of the genome by their elevated CpG dinucleotide density and the fact that they evade DNA methylation, which would otherwise be repressive to transcription[7,8]. CGIs are bound by proteins that interact specifically with non-methylated CpG dinucleotides and these CGI-reading proteins tend to be enriched downstream of transcription start sites (TSSs) where CpG density is highest within the CGI[9–15]. Most CGI-binding proteins are part of chromatin-modifying complexes that regulate post-translational modifications on histones and create chromatin states that regulate gene expression[16,17]. For example, we and others have shown that the SET1 complexes, which contain the SET1A or SET1B histone methyltransferases[18–23], associate with CGIs via the non-methylated CpG-binding protein CFP1, whilst additional multivalent interactions with chromatin and the transcriptional machinery

[1]Department of Biochemistry, University of Oxford, Oxford, UK. ✉e-mail: rob.klose@bioch.ox.ac.uk

lead to their specific enrichment at actively transcribed genes[12,24–30]. At these sites, SET1 complexes contribute to the deposition of histone H3 lysine 4 methylation (H3K4me3), which both reinforces SET1 complex binding and enables the occupancy of additional reader proteins that have been proposed to further modify histones, remodel nucleosomes, and directly influence RNA Pol II activity to support gene expression[30–40].

Based on their proposed role in depositing H3K4me3 at actively transcribed genes, SET1 complexes are generally considered transcriptional activators. However, counterintuitively, disruption of SET1 complexes often causes both increases and decreases in gene expression that do not necessarily correlate with alterations in H3K4me3 at affected genes[12,24,30,40–50]. Furthermore, cells lacking the methyltransferase domain of SET1A are viable, yet complete removal of SET1A causes cell and early embryonic lethality, suggesting that SET1 complexes may also regulate gene expression independently of histone methylation[39,47,51]. In line with this possibility, SET1 complexes contain a protein called WDR82 that directly interacts with RNA Pol II and could provide an alternative mechanism to influence RNA Pol II activity and transcription[22,52,53]. However, further complicating matters, WDR82 has documented roles in transcription termination at the 3′ end of genes and SET1 proteins have also been proposed to be involved in termination[42,54,55]. Therefore, despite decades of intense study on SET1 complexes and emerging evidence implicating them in human disease[56–59], how they regulate gene expression remains unclear.

Additional systems that function independently of chromatin to regulate RNA Pol II elongation also have central roles in controlling transcription and gene expression[2,3]. For example, at most genes there is a checkpoint ~30–50 base pairs downstream of the TSS where initiated RNA Pol II pauses. Pausing is overcome by mechanisms that promote the release of RNA Pol II into productive elongation[3]. However, a large fraction of paused RNA Pol II does not continue into productive elongation and instead undergoes premature transcription termination (PTT)[5,60–63]. PTT can also occur further into transcribed genes where it is often associated with stable nucleosomes at the boundary of promoter-associated CGIs, or even further into the gene at cryptic polyadenylation sites[5,64–67]. Transcripts arising from PTT events are usually subject to rapid turnover by the nuclear exosome[68]. Numerous factors can contribute to PTT, including the Integrator complex, which binds to RNA Pol II and contributes to PTT by cleaving nascent RNA as it exits the RNA Pol II active site, and the cleavage and polyadenylation (CPA) complex, which recognises cryptic polyadenylation signals to promote PTT further into the gene[64,65,67,69–77]. Recently an additional factor, ZC3H4, was shown to attenuate extragenic and long non-coding RNA transcription, resulting in transcription termination[78,79]. Importantly, ZC3H4 binds to the RNA Pol II-interacting protein WDR82 and binding of ZC3H4 to WDR82 appears to be important for its effects on transcription[19,53,78,80,81].

If uncontrolled, PTT would be highly detrimental to gene expression. Therefore, mechanisms have evolved to oppose and regulate specific PTT pathways[4,5]. For example, in addition to its role as a component of the spliceosome, the U1 snRNP can independently bind to 5′ splice sites in nascent RNA and inhibit the activity of the CPA machinery at downstream cryptic polyadenylation sites[64,65,82–84]. Furthermore, TFIIS can help to restart backtracked RNA Pol II to limit PTT, while SCAF4/8 interact with elongating RNA Pol II and suppress gene-intrinsic polyadenylation site usage[85,86]. These examples demonstrate that control of PTT can provide an additional mechanism to regulate gene expression. However, our current understanding of the factors that regulate PTT and the molecular logic underpinning how these systems control gene expression remains rudimentary and is a major conceptual gap in our understanding of gene regulation.

Here we set out to understand how the CpG-binding and chromatin-modifying SET1 complexes regulate gene expression. Using genome engineering, degron approaches, and quantitative genomics, we demonstrate that despite SET1 complexes binding to the majority of actively transcribed CGI-associated gene promoters, they play a very specific role in enabling the expression of low to moderately transcribed genes. This effect does not rely on their histone methyltransferase activity, nor is it related to effects on H3K4me3, but instead SET1 complexes regulate gene expression by interacting with the RNA Pol II-binding protein WDR82. Unexpectedly, we discover that SET1 complexes enable gene expression by specifically antagonising PTT by the ZC3H4/WDR82 complex, which we show pervasively terminates both genic and extra-genic transcription. As such, we uncover a gene regulatory logic whereby a CGI-binding complex is enriched on CpG-rich DNA downstream of active TSSs to ensure low to moderately transcribed genes are protected from PTT, thus limiting the activity of a pervasive PTT mechanism to extra-genic transcription.

## Results
### SET1 complexes primarily enable expression of low to moderately expressed genes
Despite their intimate association with actively transcribed CGI-associated gene promoters, it remains very poorly understood whether SET1 complexes play a significant role in regulating gene expression[30]. Addressing this important question has been extremely challenging given that SET1 complexes are essential for cell viability and traditional perturbation approaches previously used to study their function are slow. This means existing gene expression analyses will be confounded by pleiotropic secondary effects that inevitably result from deteriorating cell viability. To overcome this limitation, we used a rapid degron approach and quantitative time-resolved genomics to examine how SET1 complexes regulate gene expression in mouse embryonic stem cells (ESCs)[87,88]. We focussed initially on SET1A as it is essential in ESCs and is proposed to contribute centrally to H3K4me3 deposition at actively transcribed genes[39,40]. We used CRISPR/Cas9-mediated genome engineering to introduce a degradation tag (dTAG) into the *Set1a* gene[87]. The addition of the dTAG did not affect SET1A protein levels (Fig. 1a), disrupt SET1A complex formation (Supplementary Fig. 1a), or affect SET1A localisation to CpG-rich regions downstream of TSSs at expressed CGI-associated gene promoters (Fig. 1c, d and Supplementary Fig. 1c). Importantly, within 2 h of treatment with dTAG13, there was a near-complete loss of SET1A protein and its occupancy at target genes as assessed by western blot and calibrated chromatin immunoprecipitation coupled to sequencing (cChIP-seq) (Fig. 1a, c, d and Supplementary Fig. 1b).

Having shown that we can rapidly deplete SET1A, we then examined gene expression using calibrated total RNA-seq (cRNA-seq) at several time points after SET1A removal (Fig. 1b and Supplementary Fig. 1f). Remarkably, after only two hours of SET1A depletion we observed profound effects on gene expression and, in contrast to previous findings[12,24,30,40–47], we found that the removal of SET1A predominantly resulted in reductions in gene expression (2299 genes) with a much smaller number of genes increasing in expression (414). Importantly, these effects were due to depletion of SET1A, as treating wild type cells with dTAG13 caused no significant alterations in gene expression (Supplementary Fig. 1d). Interestingly, changes in gene expression were less pronounced at later time points after SET1A depletion, suggesting that additional mechanisms or cellular adaptations may compensate for its depletion over time (Supplementary Fig. 1f). This highlights the importance of using rapid depletion to capture primary effects on gene expression when studying essential proteins.

Although SET1A is highly expressed and thought to predominate in forming the SET1 complex in ESCs, its paralogue SET1B is also expressed (Fig. 1e). Furthermore, whilst the majority of genes were reduced in expression following depletion of SET1A, *Set1b* expression was increased (Supplementary Fig. 1e). This suggested that SET1B

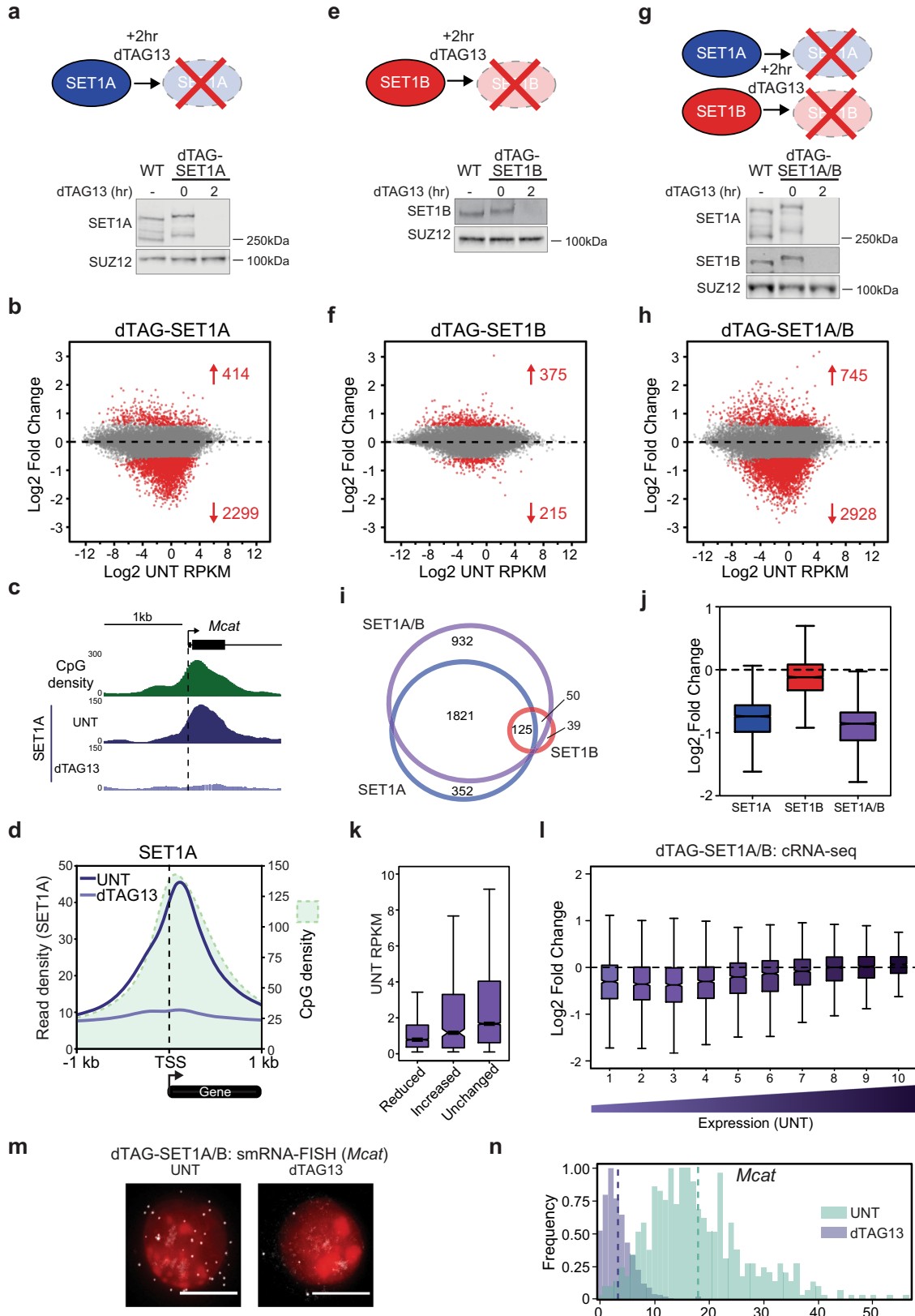

**c** Mcat

**d** SET1A

**i**

**j**

**k**

**l** dTAG-SET1A/B: cRNA-seq

**m** dTAG-SET1A/B: smRNA-FISH (*Mcat*)

**n** *Mcat*

might also regulate gene expression and possibly have a compensatory role after SET1A depletion. To examine this possibility and to ensure we had removed all SET1 complex activity, we created cell lines in which we introduced a dTAG into the *Set1b* gene, or into both *Set1a* and *Set1b* genes (Fig. 1e, g). In contrast to SET1A depletion, after 2 h of SET1B depletion there were only minimal effects on gene expression,

which were almost completely absent at later time points (Fig. 1f and Supplementary Fig. 1g). However, when SET1A and SET1B were depleted simultaneously there was an even more pronounced reduction in gene expression than was observed following depletion of SET1A alone (Fig. 1h and Supplementary Fig. 1h). Now, 2928 genes were significantly reduced in expression, whereas far fewer genes (745) were

**Fig. 1 | SET1 complexes primarily enable expression of low to moderately expressed genes. a** A schematic illustrating the approach used to rapidly deplete SET1A (top panel). A representative western blot ($n > 3$) showing comparable SET1 levels in wild type (WT) and dTAG-SET1A lines (bottom panel), and that 2 h of dTAG13 treatment causes depletion of dTAG-SET1A. SUZ12 functions as a loading control. **b** An MA-plot showing log2 fold changes in cRNA-seq signal in the dTAG-SET1A line following 2 h of dTAG13 treatment ($n = 20633$), determined using DESeq2. Significant gene expression changes ($p$-adj < 0.05 and >1.5-fold) are coloured red and the number of significantly changed genes is indicated. **c** A genomic snapshot illustrating the preferential localisation of SET1A to the CpG-rich region downstream of the TSS at the expressed CGI-associated promoter of the *Mcat* gene. SET1A signal in untreated cells (UNT) and following 2 h of dTAG13 treatment is shown. **d** Metaplot analysis of SET1A binding (cChIP-seq) at CGI-associated TSSs in the T7-dTAG-SET1A line in cells that are either untreated (UNT, dark blue) or treated with dTAG13 for 2 h (light blue). Only TSSs that do not have a divergent TSS within 1 kb were analysed ($n = 11,930$). CpG density is shown by light green shading. **e, f** As per **a, b** but for the dTAG-SET1B line. **g, h** As per **a, b** but for the dTAG-SET1A/B line. **i** A Venn diagram showing the overlap between genes with a significant reduction in expression in the dTAG-SET1A, dTAG-SET1B, and dTAG-SET1A/B lines. **j** A box plot showing the log2 fold changes in cRNA-seq signal in each

of the dTAG cell lines for the complete set of genes that rely on SET1 complexes for their expression ($n = 3320$). The boxes show interquartile range, centre line represents median, whiskers extend by 1.5× IQR or the most extreme point (whichever is closer to the median), while notches extend by 1.58× IQR/sqrt(n), giving a roughly 95% confidence interval for comparing medians. **k** A box plot showing the expression level in untreated cells (UNT RPKM) for expressed genes with reduced expression following 2 h of SET1A/B depletion (Reduced, $n = 2544$), increased expression (Increased, $n = 495$) and unchanged expression (Unchanged, $n = 9989$). Boxes are defined as in **j. l** A box pot showing the log2 fold change in cRNA-seq signal in the dTAG-SET1A/B line after 2 h dTAG13 treatment with expressed genes ($n = 13,028$) separated into deciles based on their expression level in untreated cells. Boxes are defined as in **j. m** Example images of smRNA-FISH for the *Mcat* gene in the dTAG-SET1A/B line, showing an untreated (UNT) cell and a cell treated with dTAG13 for 2 h. White corresponds to *Mcat* RNAs and red corresponds to DAPI staining of DNA. The white scale bars correspond to 10 μm. **n** A histogram illustrating the number of transcripts per cell as measured by smRNA-FISH before (UNT, light green) and after 2 h of dTAG13 treatment (light purple) for the *Mcat* gene in the dTAG-SET1A/B line. The dashed lines correspond to the mean of the distribution.

increased in expression and these effects were much smaller in magnitude. This suggests that SET1A and SET1B both contribute to gene regulation. This was also evident when we examined the overlap in the genes that rely on SET1 proteins for their expression, where it was clear that reductions in expression were largest in magnitude when SET1A and SET1B were simultaneously depleted (Fig. 1i, j). Importantly, genes that rely on SET1 proteins for their expression almost all had a promoter-associated CGI (Supplementary Fig. 1k). However, despite SET1 proteins being enriched at expressed CGI-associated gene promoters, only a subset (~20%) require SET1 proteins for their expression. This suggested that genes that rely on SET1 proteins may have some shared feature that renders them sensitive to SET1 protein depletion.

To examine this possibility in more detail, we first asked whether genes that rely on SET1 proteins for their expression might differ in their CGI attributes from those that do not. Interestingly, this analysis showed that effects on gene expression were not related to the size of the CGI, its CpG density, SET1A occupancy, nor H3K4me3 levels (Supplementary Fig. 1i). Furthermore, we found no specific ontology terms amongst genes with reduced expression after SET1A/B depletion, suggesting that there was no defined type of gene that relies on SET1A/B for expression (Supplementary Fig. 1j). Instead, the only obvious feature distinguishing genes that rely on SET1 proteins for their expression was that they tended to be more lowly expressed than unchanged genes (Fig. 1k). When we examined the relationship between expression level and sensitivity to SET1 protein depletion in more detail, it was clear that low to moderately expressed genes were the most sensitive, with more highly expressed genes not relying on SET1 proteins for their expression (Fig. 1l). Therefore, despite SET1 proteins binding to most active CGI-associated gene promoters, our rapid depletion approaches reveal that SET1A and SET1B play a prominent and overlapping role in supporting gene expression at low to moderately expressed genes.

cRNA-seq analysis measures average changes in gene expression across millions of cells and therefore does not capture how these effects ultimately manifest in individual cells within the population. To address this important question we carried out single molecule RNA fluorescent in situ hybridisation (smRNA-FISH) to enable absolute quantification of gene expression changes in single cells for three SET1-dependent genes (Fig. 1m, n and Supplementary Fig. 1l, m)[89]. Importantly, for each of the genes examined, the reductions in gene expression after SET1A/B depletion were on average uniform across the cell population. Therefore, our genomic and imaging analysis reveal that SET1 complexes have a widespread, overlapping, and uniform role in enabling the expression of low to moderately expressed genes.

## SET1 complexes can regulate gene expression independently of H3K4me3 and methyltransferase activity

SET1 complexes are thought to be the predominant H3K4 tri-methyltransferases in animals and the deposition of H3K4me3 has been proposed to influence gene expression[30,39–41,90–92]. As our degron approach allows us to capture the earliest and most primary influences of SET1 protein depletion on gene expression, we set out to examine whether the observed effects might result from changes in H3K4me3. Initially we examined the bulk levels of H3K4me3 by western blot after SET1 protein depletion (Fig. 2a and Supplementary Fig. 2a). This revealed only a very modest and non-significant reduction in H3K4me3, even after several days of SET1 protein depletion. However, bulk western blot analysis does not allow us to examine gene-specific effects on H3K4me3, which could be more closely related to effects on gene expression. Therefore, we also carried out cChIP-seq for H3K4me3 at 2, 4, and 24 h after SET1 protein depletion (Fig. 2b, c and Supplementary Fig. 2b–d). Again, this revealed only very modest reductions in H3K4me3 at gene promoters, with no significant correlation between the effects on H3K4me3 and reductions in gene expression (Fig. 2b–d and Supplementary Fig. 2b–e). This demonstrates that acute removal of SET1 complexes does not lead to major alterations in H3K4me3 at gene promoters in ESCs and suggests that the effects we observe on gene expression after their depletion may be independent of H3K4me3 and SET1 complex methyltransferase activity.

Based on these findings, we sought to more directly examine whether SET1 complexes can regulate gene expression independently of their methyltransferase activity. To achieve this, we developed a chromatinised gene reporter system containing a single copy transgene in which Tet operator DNA binding sites (TetO) were coupled to a minimal gene promoter that drives luciferase expression (Fig. 2e). Fusing a protein of interest to the reverse Tet Repressor DNA binding domain (rTetR) enables its recruitment to the TetO array upon the addition of doxycycline (Fig. 2e). Consistent with a role for SET1 complexes in supporting gene expression, tethering wild-type SET1A to the promoter resulted in increased reporter gene expression (Fig. 2f). Interestingly, tethering a version of SET1A in which we had mutated key residues required for its methyltransferase activity supported gene expression in a manner that was similar to the wild type protein (Fig. 2f and Supplementary Fig. 2f, g)[93–96]. Together, our histone modification analysis and tethering experiments suggest that alterations in gene expression observed after SET1 protein depletion do not primarily manifest from a loss of H3K4me3 and that SET1A can support reporter gene expression independently of its methyltransferase activity.

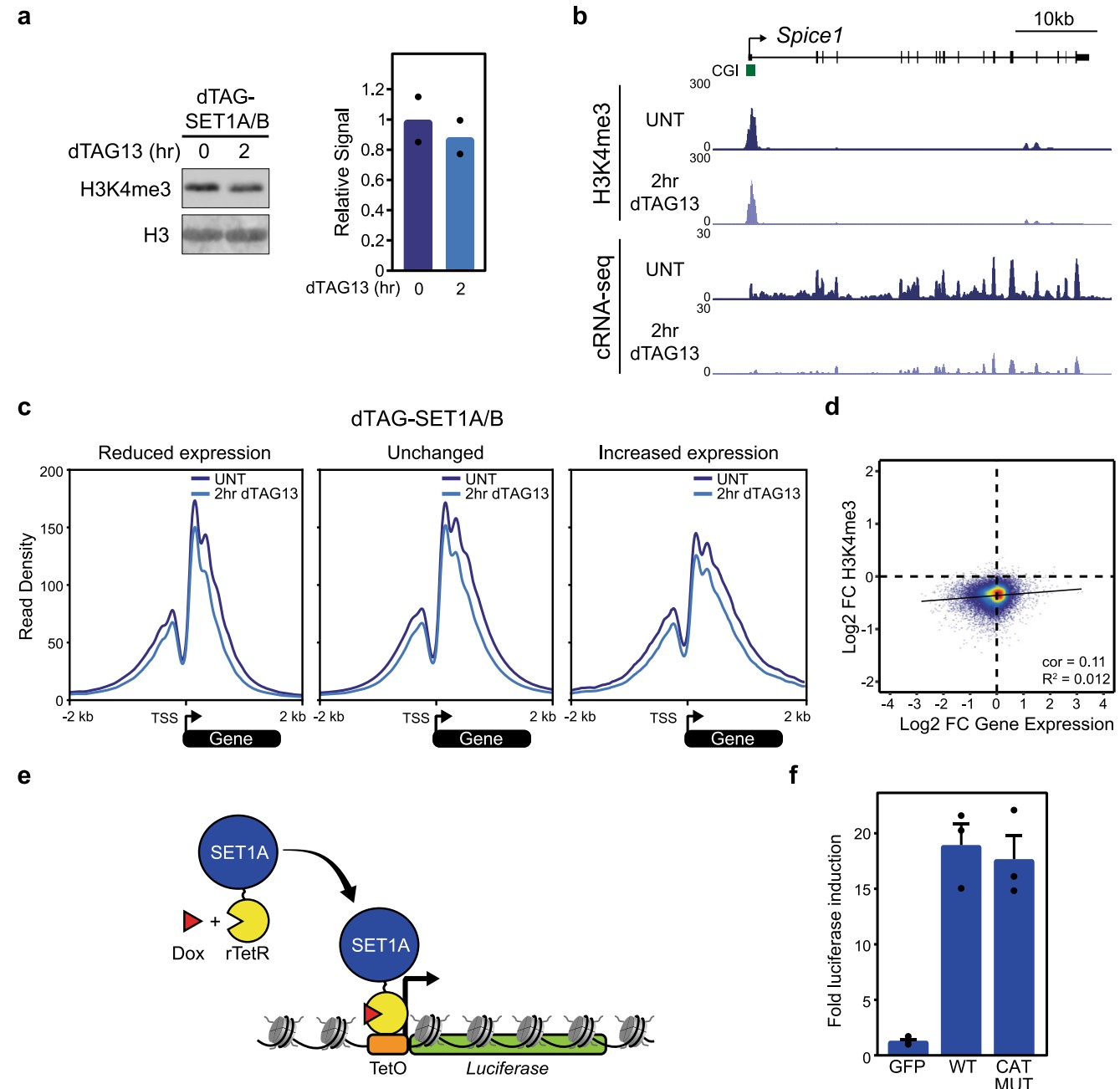

**Fig. 2 | SET1 complexes can regulate gene expression independently of H3K4me3 and methyltransferase activity. a** Western blot analysis of H3K4me3 levels in the untreated dTAG-SET1A/B line and following 2 h of dTAG13 treatment (left panel). H3 is included as a loading control. Mean H3K4me3 levels before and after dTAG13 treatment from two biological replicates is shown (right panel). **b** A genomic snapshot comparing H3K4me3 cChIP-seq signal (top panels) and cRNA-seq signal (bottom panels) before and after 2 h of dTAG13 treatment in the dTAG-SET1A/B line at the *Spice1* gene. **c** Metaplot analysis of H3K4me3 cChIP-seq around the transcription start site (TSS) of genes that have reduced expression (left panel), unchanged expression (middle panel), or increased expression (right panel) in the dTAG-SET1A/B line, before (UNT, dark blue lines) and after 2 h of dTAG13 treatment (light blue lines). Only expressed genes are included (reduced expression, $n = 2544$;

unchanged, $n = 9989$; increased expression, $n = 495$). **d** A scatter plot comparing the log2 fold change (Log2 FC) in H3K4me3 cChIP-seq signal and cRNA-seq signal (gene expression) in the dTAG-SET1A/B line following 2 h of dTAG13 treatment. The pearson correlation (cor) and $R^2$ values are indicated. Only genes that have a peak of H3K4me3 in untreated cells are included ($n = 14065$). **e** A schematic illustrating the chromatinised reporter gene. TetO binding sites are coupled to a minimal core promoter and a luciferase reporter gene. rTetR-fusion proteins are tethered to the reporter gene by the addition of doxycycline (Dox) and effects on gene expression can be monitored. **f** A bar plot showing the mean fold induction of reporter gene expression following tethering of GFP, SET1A (WT) and SET1A with catalytic mutations in its SET domain (CAT MUT). Error bars represent SEM from three biological replicates.

## SET1 complexes support gene expression through an interaction with WDR82

Given that SET1A can support gene expression independently of its methyltransferase activity, we set out to determine what region of the protein was responsible for this effect. To address this, we took advantage of our reporter gene system and tested the capacity of

various SET1A fragments to support gene expression. These fragments included the conserved N-terminal region that has been proposed to interact with WDR82[22,97]; the RRM domain which, in other proteins, can interact with RNA[98]; the central region of SET1A which lacks significant sequence conservation; and the N-SET/SET domain which interacts with CFP1/WDR5/RBBP5/ASH2L/DPY30 and is required for chromatin

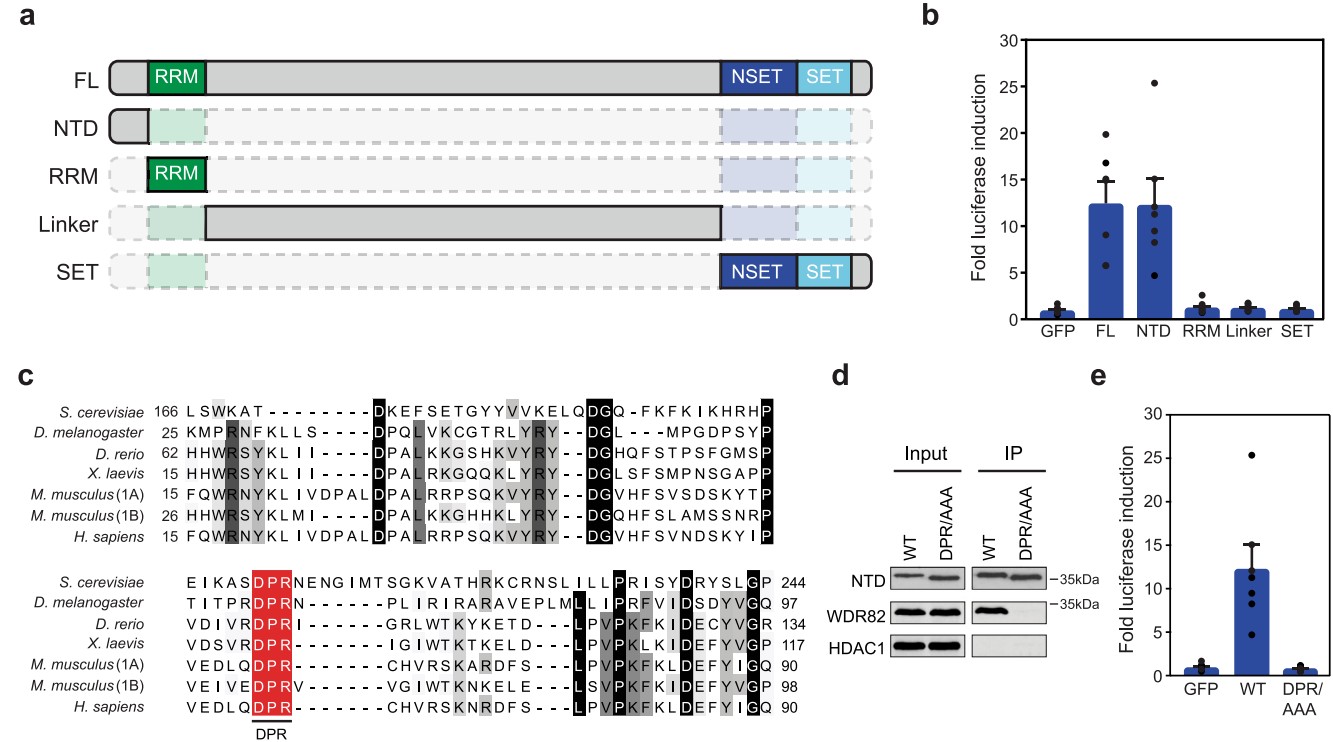

**Fig. 3 | SET1 complexes support gene expression through an interaction with WDR82. a** A schematic illustrating the SET1A domains that were tethered to the reporter gene. **b** A bar plot showing the mean fold induction of reporter gene expression following tethering of the GFP, full length SET1A (FL), NTD, RRM, Linker, and the NSET-SET (SET) domain fragments to the reporter gene. Error bars represent SEM from seven biological replicates. **c** A multiple sequence alignment of the N-terminal domains of SET1A (1A) or SET1B (1B) from the indicated species. The red box highlights the conserved and invariant DPR motif. **d** An immunoprecipitation (IP) of the NTD of WT SET1A or the DPR/AAA mutant followed by western blot for WDR82. HDAC1 functions as a loading control for the input samples and a negative control for interaction with SET1A. **e** A bar plot showing the mean fold induction of reporter gene expression following tethering of GFP, WT-NTD, or the DPR/AAA-NTD of SET1A. Error bars represent SEM from seven biological replicates.

binding and methyltransferase activity (Fig. 3a)[30]. Interestingly, only the short N-terminal domain (NTD) of SET1A was sufficient to support gene expression, and it did so to a similar extent as the full-length protein (Fig. 3b and Supplementary Fig. 3a). This suggests that the other domains in SET1A, and their interacting proteins, do not contribute significantly to supporting gene expression in this context. Furthermore, the equivalent NTD of SET1B was also sufficient to support gene expression, and it did so more efficiently than the NTD of SET1A, indicating that this activity is conserved amongst SET1 paralogues (Supplementary Fig. 3a, b).

Having shown that the NTDs of SET1 proteins are sufficient to support gene expression, we hypothesised that this effect may rely on their capacity to interact with WDR82. To test this possibility, we carried out sequence conservation analysis of the NTD across SET1 orthologues to identify a putative WDR82 interaction motif (Fig. 3c). This revealed a highly conserved trio of amino acids, herein referred to as the DPR motif. When the DPR motif was mutated to alanines (DPR/AAA), the SET1A NTD was unable to interact with WDR82 (Fig. 3d). Importantly, when we tethered the DPR/AAA-NTD of SET1A to the reporter gene promoter, it was unable to support gene expression (Fig. 3e). Therefore, a highly conserved DPR motif in the N-terminal domains of SET1 proteins is required for their interaction with WDR82 and this interaction can support gene expression either directly or through additional protein interactions.

## SET1 complexes support genic transcription downstream of the TSS

Having discovered that SET1A can regulate gene expression through binding to WDR82, and knowing that WDR82 can interact with RNA Pol II[22], we hypothesized that SET1 complexes may directly influence RNA

Pol II occupancy and/or transcription. To explore this possibility, we examined ongoing transcription using calibrated transient transcriptome sequencing (cTT-seq) and RNA Pol II occupancy using cChIP-seq (Fig. 4a).

We first carried out cTT-seq analysis to examine gene transcription after depleting SET1 proteins. This revealed that 3098 genes, corresponding to approximately 22% of transcribed genes, had reduced transcription after SET1 protein depletion, whereas only 75 genes exhibited an increase in transcription (Fig. 4b). This demonstrates that SET1 proteins almost exclusively support gene transcription. In agreement with our findings from cRNA-seq, the influence of SET1 proteins on transcription was limited to low to moderately transcribed genes (Supplementary Fig. 4a–c). However, in comparison to cRNA-seq, cTT-seq identified more genes that had reduced transcription and these reductions were larger in magnitude (Supplementary Fig. 4d). This difference likely arises from the fact that cRNA-seq interrogates total RNA levels which are influenced by both the rate of transcript production and degradation. Indeed, genes that were reduced in expression after SET1 complex depletion, as measured by cRNA-seq, tended to have shorter transcript half-lives than unchanged genes (Supplementary Fig. 4e). In contrast, genes with reduced transcription, as measured by cTT-seq, were not subject to this bias. Therefore, we conclude that cTT-seq captures the primary influences of SET1 complexes on gene transcription.

To better understand how SET1 complexes influence the function of RNA Pol II to support gene transcription, we next examined RNA Pol II occupancy by cChIP-seq. Interestingly, at both SET1-dependent and SET1-independent genes, we observed only very modest effects on RNA Pol II binding over the region corresponding to the transcription start site and promoter-proximal pause site (Fig. 4c, d and

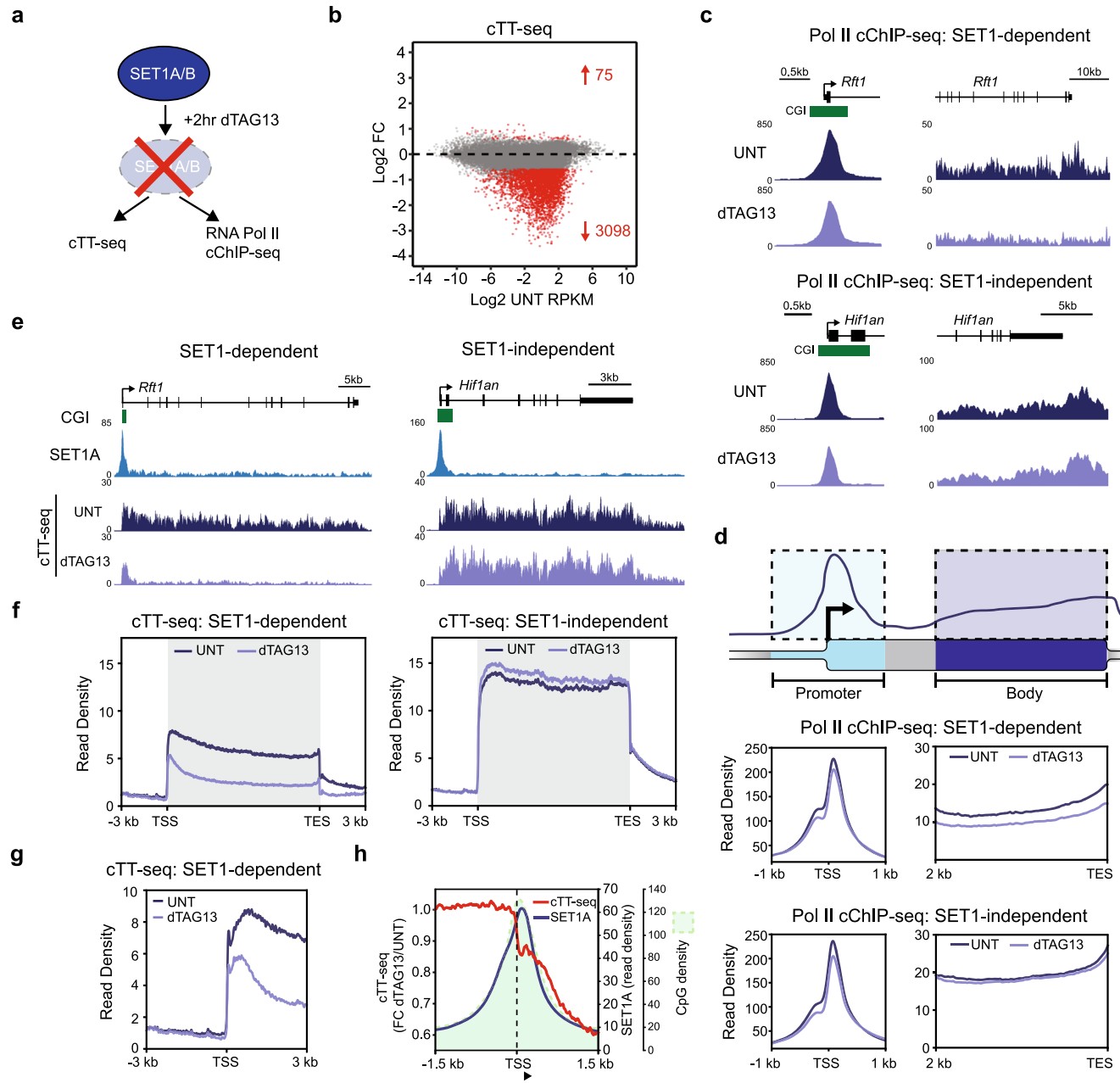

**Fig. 4 | SET1 complexes support genic transcription downstream of the TSS.**
**a** A schematic illustrating the experiments carried out in the dTAG-SET1A/B line to examine RNA Pol II occupancy (RNA Pol II cChIP-seq) and transcription (cTT-seq).
**b** An MA-plot showing log2 fold changes (Log2 FC) in transcription (cTT-seq) in the dTAG-SET1A/B line following 2 h of dTAG13 treatment ($n = 20633$), determined using DESeq2. Significant changes in transcription (p-adj < 0.05 and >1.5-fold) are coloured red and the number of significantly changed genes is indicated.
**c** Genomic snapshots of RNA Pol II occupancy (Pol II cChIP-seq) at a SET1-dependent (upper panel, *Rft1*) and SET1-independent (lower panel, *Hif1an*) gene in untreated cells (UNT, dark purple) or cells treated with dTAG13 for 2 h (light purple). CGIs are shown in green. The left hand panels correspond to gene promoter occupancy and the right hand panels to gene body occupancy. **d** The top panel is a cartoon schematic illustrating the typical RNA Pol II cChIP-seq signal over a gene with the gene promoter region highlighted in light blue and the gene body in light purple. The bottom panels correspond to metaplot analysis of RNA Pol II cChIP-seq signal in the dTAG-SET1A/B line in cells that are either untreated (UNT) or treated with dTAG13 for 2 h. The RNA Pol II cChIP-seq signal corresponding to the gene promoter and body regions (see schematic in top panel) of all transcribed SET1-

dependent genes (middle panel, $n = 2633$) and SET1-independent genes (bottom panel, $n = 9151$) is shown. **e** Genomic snapshots of cTT-seq signal in the dTAG-SET1A/B line at a SET1-dependent (left panel, *Rft1*) and SET1-independent (right panel, *Hif1an*) gene in untreated cells (UNT, dark purple) or cells treated with dTAG13 for 2 h (light purple). The CGIs are shown in green and SET1A cChIP-seq signal in light blue. **f** Metaplot analysis of transcription (cTT-seq) in the dTAG-SET1A/B line in cells that are either untreated (UNT, dark purple) or treated with dTAG13 for 2 h (light purple) for all actively transcribed SET1-dependent genes (left panel, $n = 2633$) and SET1-independent genes (right panel, $n = 9151$). **g** Metaplot analysis of transcription (cTT-seq) in the dTAG-SET1A/B line in cells that are either untreated (UNT, dark purple) or treated with dTAG13 for 2 h (light purple) zoomed in to the transcription start site (TSS) of all actively transcribed SET1-dependent genes ($n = 2633$). **h** Metaplot analysis of the fold change in cTT-seq signal (red line) between dTAG13-treated and untreated dTAG-SET1A/B cells at the transcription start site (TSS) of all transcribed SET1-dependent genes ($n = 2633$). Following SET1A depletion, the attenuation of transcription occurs downstream of TSSs over the CpG-rich region (green shaded area) of the CGI, coincident with the location of SET1A binding in untreated cells (dark blue line).

Supplementary Fig. 4f, g). This suggests that SET1 complexes do not play a central role in influencing RNA Pol II occupancy during the very earliest stages of transcription. However, we did observe reductions in RNA Pol II occupancy in the body of SET1-dependent genes, an effect that was not evident at SET1-independent genes (Fig. 4c, d and Supplementary Fig. 4f, g). This suggests that SET1 complexes, through binding to CpG-rich DNA downstream of TSSs, may support RNA Pol II in transcribing into the gene body.

To investigate this possibility in more detail, we further examined our cTT-seq data, which also provides spatial information about the level of transcription across genes[99,100]. When SET1 proteins were depleted, SET1-dependent genes still initiated transcription from the gene promoter, albeit at reduced levels (Fig. 4e–h and Supplementary Fig. 4h, i). However, following initiation, transcription was attenuated downstream of TSSs in a region coincident with the CpG-rich region of the CGI where SET1 complexes bind (Fig. 4g, h). Given that we did not observe an accumulation of RNA Pol II in the promoter region or the body of SET1-dependent genes after SET1 protein depletion, this reduction in transcription does not appear to be due to increased promoter–proximal pausing or reduced elongation rate in the gene body (Fig. 4d)[101]. Therefore, we propose that the observed attenuation of transcription in the absence of SET1 complexes may be caused by premature transcription termination (PTT), which would be consistent with largely unaffected promoter-proximal occupancy of RNA Pol II and a reduction of RNA Pol II in the body of SET1-dependent genes. PTT might also in part explain the reduced cTT-seq signal observed at the 5′ end of SET1-dependent genes following SET1 protein depletion, since the products of PTT are often rapidly degraded by the exosome[5,68]. Together these observations suggest that SET1 complexes function to counteract transcription termination at CGIs, and despite broadly associating with CGIs and localising to the CpG-dense region downstream of most actively transcribed TSSs, this is particularly important for the transcription of low to moderately transcribed genes.

## ZC3H4 contributes to the termination of extragenic and genic transcription

It has recently been demonstrated that PTT pathways play an important role in controlling gene expression[102]. Based on the observation that SET1 complexes appear to counteract transcription attenuation, we reasoned that they may do so by antagonising an opposing PTT activity. We have shown that SET1 complexes can support gene expression via an interaction with the RNA Pol II-binding protein, WDR82. Interestingly, WDR82 also interacts with ZC3H4, which has recently been shown to localise with RNA Pol II and contribute to transcription termination, particularly of extragenic transcription[19,53,78–81,102]. Therefore, we hypothesised that the enrichment of WDR82-containing SET1 complexes may support transcription by antagonising PTT by WDR82-containing ZC3H4 complexes.

To examine this possibility, we set out to explore how ZC3H4 influences gene transcription. To achieve this, we first epitope-tagged the endogenous *Zc3h4* gene and carried out ChIP-seq analysis to examine ZC3H4 binding in the genome (Fig. 5a, b and Supplementary Fig. 5a). In contrast to SET1 proteins, and consistent with its proposed role in terminating extragenic transcription, ZC3H4 localises to active enhancers that are bound by RNA Pol II but which are not typically associated with CGIs (Fig. 5a, b and Supplementary Fig. 5b, c). As reported previously, we also found that ZC3H4 localises with RNA Pol II at actively transcribed gene promoters[78,79,81], where it is enriched on the shoulders of RNA Pol II peaks corresponding to where sense and antisense early transcription elongation complexes predominate (Fig. 5a, b). Therefore, despite previous reports that ZC3H4 primarily affects extragenic and non-coding RNAs, we reasoned that ZC3H4 might also contribute to PTT of protein-coding transcription. Importantly, while ZC3H4 enrichment was similar at both SET1-independent

and -dependent genes, it was very slightly biased downstream of TSSs towards the gene body at SET1-dependent genes, suggesting that ZC3H4 might influence SET1-dependent and -independent genes differently (Fig. 5c).

To examine the role of ZC3H4 in regulating gene transcription, we engineered a dTAG into the endogenous *Zc3h4* gene. dTAG13 treatment resulted in a near-complete depletion of ZC3H4 within 2 h, with no effect on the levels of its interaction partner WDR82 (Fig. 5d and Supplementary Fig. 5d). To examine whether ZC3H4 depletion affects transcription, we next carried out cTT-seq. In the absence of ZC3H4, upstream antisense transcription from promoters and enhancer transcription were both increased, consistent with the proposed role for ZC3H4 in terminating extragenic transcription in other cell types (Fig. 5e)[78,79,81]. In contrast, transcription at these regions was not significantly affected by SET1 depletion, consistent with SET1 proteins being largely absent from enhancers and primarily affecting genic transcription (Fig. 5f).

Strikingly, when we analysed genic transcription after ZC3H4 depletion, we observed increased transcription of 2599 genes (Fig. 5g, h). This indicates that ZC3H4 also significantly counteracts genic transcription, although these effects were of lesser magnitude than at extragenic regions (Supplementary Fig. 5e). Interestingly, genes that were most susceptible to ZC3H4 loss were typically low to moderately transcribed, and had low CpG density and SET1A occupancy (Fig. 5i, j). We presume that highly transcribed genes, despite also being bound by ZC3H4, must have sufficient activation signals and other transcription processivity influences that the inhibitory effect of ZC3H4 on transcription output is nominal. Therefore, we propose that ZC3H4 associates broadly with transcribing RNA Pol II, where it contributes to transcription termination and primarily influences the transcriptional output from low to moderately transcribed regions, including lowly expressed genes and non-coding extragenic loci. This is consistent with recent suggestions that ZC3H4 complexes might act as a transcriptional surveillance mechanism to limit low level or non-productive transcription[102].

## SET1 complexes counteract premature transcription termination by ZC3H4

A shared feature amongst genes regulated by ZC3H4 and SET1 proteins is that they tend to be low to moderately transcribed (Figs. 4, 5). Therefore, we were curious whether ZC3H4 regulates the transcription of SET1-dependent genes. Consistent with this possibility, after depletion of ZC3H4 we observed a small increase in transcription at SET1-dependent genes (Fig. 6a, b, Supplementary Fig. 6a, e). Interestingly, SET1-dependent genes with significantly increased transcription after ZC3H4 depletion were more lowly transcribed than other SET1-dependent genes, whereas the genes that were only affected after ZC3H4 depletion had even lower levels of transcription (Supplementary Fig. 6b). These observations are consistent with the idea that ZC3H4 functions pervasively throughout the genome to drive PTT at lowly transcribed regions and that this activity can influence SET1-dependent genes. Furthermore, if SET1 proteins oppose ZC3H4-dependent PTT as we hypothesise, this would imply that SET1 activity must not be sufficient to completely counteract the influence of ZC3H4, particularly at the most lowly transcribed genes.

Having demonstrated that ZC3H4 can influence SET1-dependent genes, we next set out to directly test whether SET1 complexes enable gene transcription by antagonising ZC3H4-dependent PTT. To address this important question, we engineered a dTAG into the *Zc3h4* gene in the dTAG-SET1A/B cell line (Supplementary Fig. 6c). If SET1 complexes primarily function to antagonise ZC3H4-dependent PTT, we would expect the profound attenuation of transcription observed when SET1 proteins are depleted (Fig. 6c, d and Supplementary Fig. 6f) to be completely ameliorated by the simultaneous depletion of both SET1 proteins and ZC3H4, with the resulting cTT-seq signal resembling that

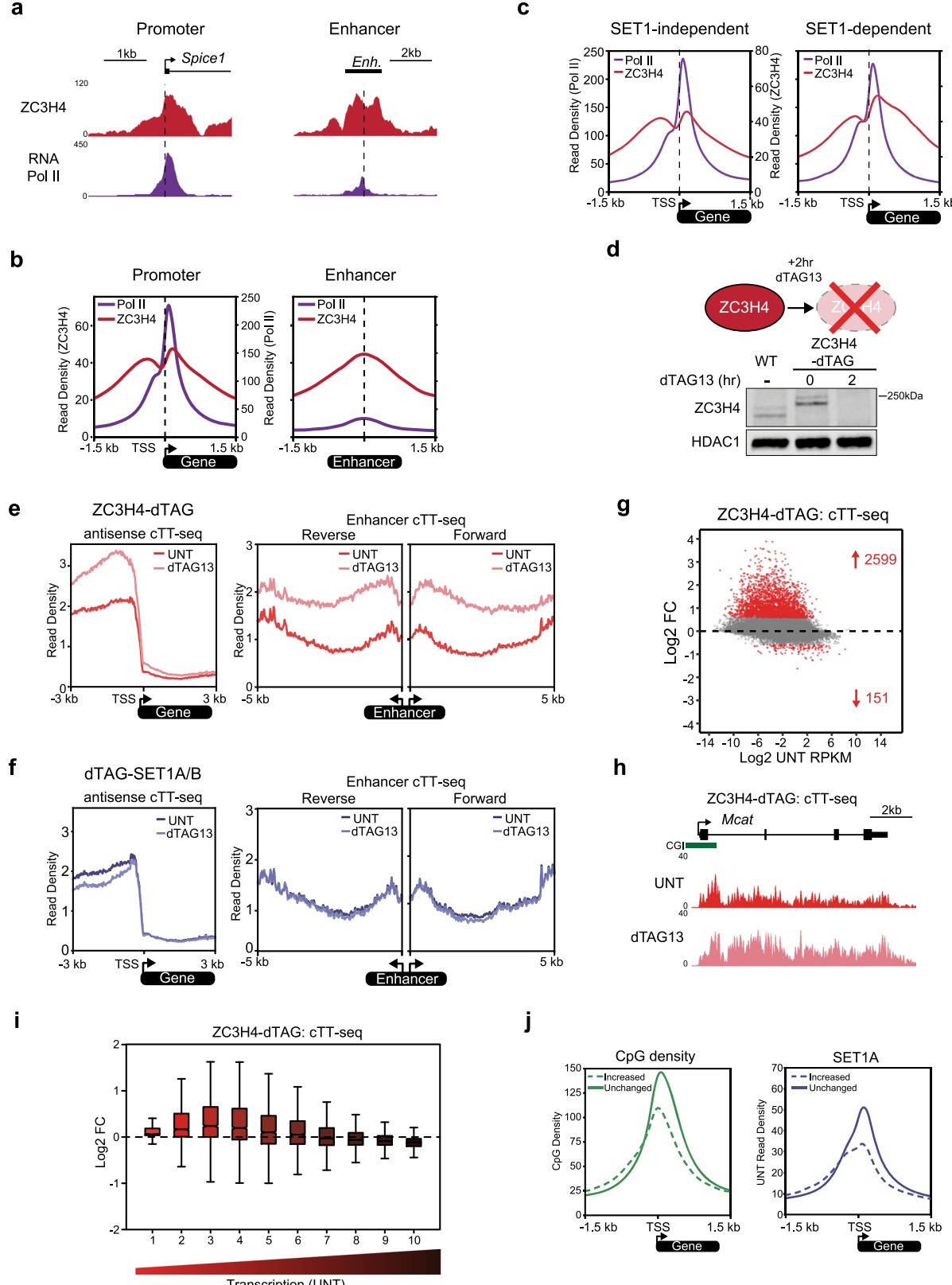

observed when only ZC3H4 is depleted. To test this, we treated the dTAG-SET1A/B/ZC3H4 line with dTAG13 to simultaneously deplete both SET1 proteins and ZC3H4 and carried out cTT-seq (Fig. 6e, f, Supplementary Fig. 6d, g). Remarkably, this revealed that the attenuation of transcription caused by depletion of SET1 proteins was completely reversed when ZC3H4 was simultaneously depleted and the cTT-seq signal now mirrored that of ZC3H4 removal alone (Fig. 6a–g,

Supplementary Fig. 6d–g). These findings are consistent with the idea that SET1 complexes primarily function to antagonise PTT by ZC3H4 complexes. Importantly, this antagonism does not appear to manifest from direct physical competition for binding to RNA Pol II or WDR82, as depletion of SET1 complexes did not lead to increased binding of ZC3H4 at TSSs and WDR82 is in excess to SET1 and ZC3H4 complexes in ESCs (Supplementary Fig. 6h, i). As such, we envisage that both SET1 and

**Fig. 5 | ZC3H4 terminates low to moderately transcribed extragenic and genic transcription. a** Genomic snapshots illustrating ZC3H4 and RNA Pol II ChIP-seq signal at the promoter of the *Spice1* gene (left panel) and an intergenic enhancer region (chr16:30,818,164-30,824,484) (right panel). **b** Metaplot analysis of ZC3H4 and RNA Pol II ChIP-seq at the TSSs of transcribed genes (*n* = 11,823, left panel) and enhancers (*n* = 4156, right panel). The read density of ZC3H4 is shown on the left axis and RNA Pol II is shown on the right axis. **c** Metaplot analysis of RNA Pol II and ZC3H4 ChIP-seq at the TSSs of transcribed SET1-independent (*n* = 9151) and SET1-dependent genes (*n* = 2633). The read density of RNA Pol II is shown on the left axis and read density of ZC3H4 on the right axis. **d** A schematic illustrating the ZC3H4-dTAG line and a representative western blot (*n* = 3) comparing ZC3H4 levels in wild type (WT) cells and the ZC3H4-dTAG line before (0 h) and following 2 h treatment with dTAG13. HDAC1 functions as a loading control. **e** Metaplot analysis of transcription (cTT-seq) in the ZC3H4-dTAG line that is either untreated (UNT, dark red) or treated with dTAG13 for 2 h (light red), showing upstream antisense transcription at all TSSs (left panel, *n* = 20,633) and enhancer transcription (right panels,

*n* = 4156). **f** As in **e**, but for the dTAG-SET1A/B cell line. **g** An MA-plot showing log2 fold changes (Log2 FC) in transcription (cTT-seq) in the ZC3H4-dTAG line following 2 h treatment with dTAG13 (*n* = 20,633), determined using DESeq2. Significant changes in transcription (*p*-adj < 0.05 and >1.5-fold) are coloured red and the number of significantly changed genes is indicated. **h** A genomic snapshot of cTT-seq signal in the ZC3H4-dTAG line at the *Mcat* gene in untreated cells (UNT) or cells treated with dTAG13 for 2 h. **i** A box pot showing log2 fold change (Log2 FC) in transcription (cTT-seq) in the ZC3H4-dTAG line after 2 h dTAG13 treatment with all genes (*n* = 20,633) separated into deciles based on their transcription level in untreated cells. The boxes show interquartile range, centre line represents median, whiskers extend by 1.5× IQR or the most extreme point (whichever is closer to the median), while notches extend by 1.58× IQR/sqrt(n), giving a roughly 95% confidence interval for comparing medians. **j** Metaplots comparing CpG density and SET1A levels at the TSSs of CGI-associated genes that are increased in transcription after 2 h of ZC3H4 depletion (*n* = 1653) and those that are unchanged (*n* = 12,667).

ZC3H4 complexes interact with the CTD of RNA Pol II and the integration of their distinct activities determines the effect on transcription (Fig. 6h). Therefore, we propose that ZC3H4 complexes survey RNA Pol II and contribute to early termination, influencing both genic and extragenic transcription. However, the binding of SET1 complexes at CpG-rich regions downstream of actively transcribed TSSs distinguishes genic from non-genic transcription and hence protects genes with low to moderate levels of transcription from ZC3H4-dependent termination to ensure normal protein-coding gene expression (Fig. 6h).

## Discussion

The mechanisms by which CGIs regulate transcription have remained an enigmatic component of vertebrate gene regulation. Furthermore, while PTT has recently emerged as an important regulator of gene transcription, the mechanisms through which it is controlled remain poorly understood and represent a major conceptual gap in our understanding of transcription and gene regulation. Here we discover that the CGI-binding SET1 complexes primarily function to enable gene expression (Fig. 1). Unexpectedly, this can occur independently of their H3K4me3 methyltransferase activity (Fig. 2), but relies on their capacity to interact with the RNA Pol II-binding protein WDR82 (Fig. 3). We discover that removing SET1 complexes causes low to moderately transcribed genes to become acutely sensitive to PTT downstream of TSSs where SET1 complexes bind (Fig. 4). Furthermore, we reveal that PTT at low to moderately transcribed regions is driven by ZC3H4 complexes, which also contain WDR82 (Fig. 5). This suggests that SET1 complexes could function downstream of genic TSSs via their WDR82 component to counteract PTT by ZC3H4 complexes. In agreement with this, simultaneous removal of SET1 and ZC3H4 complexes reverses the requirement for SET1 complexes in gene transcription, demonstrating that SET1 complexes function at CGIs to antagonise ZC3H4-dependent PTT (Fig. 6). Therefore, we uncover an unexpected gene regulatory mechanism whereby a CGI-binding complex functions downstream of TSSs to counteract PTT and enable gene expression.

In contextualising these discoveries, it is important to consider how ZC3H4 and SET1 complexes interface with genes and extragenic regions of the genome, and the logic that might underpin how their functional integration regulates transcription. Key to this is likely the fact that ZC3H4 and SET1 complexes have a shared interaction partner, WDR82[19,53]. WDR82 preferentially binds the C-terminal heptapeptide repeat (CTD) of the largest subunit of RNA Pol II when it is phosphorylated on serine 5 (Ser5P), which occurs when initiated RNA Pol II transitions into elongation[22,52]. Although we currently have a limited understanding of the precise mechanisms that enable ZC3H4 complex targeting, in addition to binding CTD-Ser5P via WDR82[81], ZC3H4 can also interact with the nuclear RNA cap-binding complex protein ARS2[103]. Based on these interactions, we propose that ZC3H4 complexes may generically recognise early elongating RNA Pol II through

binding to CTD-Ser5P and an exposed and capped RNA. In agreement with this possibility, we and others have found that ZC3H4 co-localises with RNA Pol II at regions of transcription initiation, including both promoters and enhancers (Fig. 5). Despite this pervasive localisation with RNA Pol II, highly transcribed genes appear to be refractory to the effects of ZC3H4 and instead ZC3H4 primarily influences regions with low levels of transcription, including low to moderately transcribed genes and regions of extragenic transcription.

We envisage that the specificity of ZC3H4 complexes for lowly transcribed regions may have primarily evolved to counteract extragenic transcription, which does not usually produce functional transcripts[78,79]. Such a generic termination activity would also ensure that extragenic transcription does not have other deleterious effects on genes, including transcription interference. However, a pervasive mechanism to terminate lowly transcribed extragenic regions would also impinge on lowly transcribed genes and this could be highly detrimental to their expression. Importantly, we have shown that SET1 complex binding downstream of TSSs at actively transcribed genes specifically counteracts PTT by ZC3H4 complexes (Fig. 6). Like ZC3H4 complexes, SET1 complexes interact with WDR82, and in reporter gene experiments this interaction appears to be important for the effects of SET1A on gene expression (Fig. 3). We cannot rule out that the influence of SET1A/B on reporter gene expression is due to effects beyond counteracting transcription termination. However, based on our genome-wide studies, we propose that the binding of WDR82-containing SET1 complexes can antagonise the activity of ZC3H4/WDR82 complexes, with this being particularly important for the transcription and expression of low to moderately transcribed genes. Furthermore, we discover that SET1 complex function is required for the normal transcription of ~22% of expressed genes, which could explain their requirement for cell viability and possibly even why subtle perturbations to their function in humans leads to disease.

SET1 complexes generically associate with CpG-rich DNA just downstream of TSSs on the genic side of transcribed CGI-associated genes through multivalent chromatin binding mechanisms and may also initially sense transcription at these sites via WDR82 binding to Ser5P on the RNA Pol II CTD[12,24–30]. This binding polarity with respect to the TSS effectively distinguishes genic transcription from sites of extragenic transcription, including promoter upstream antisense transcription, which are not typically enriched for SET1 complex occupancy. Interestingly, despite pervasive localisation of SET1 complexes to transcribed genes, we discover that they primarily support the transcription of low to moderately transcribed genes and we show that this is due to their role in antagonising PTT by ZC3H4 complexes. We envisage that highly expressed genes are less susceptible to ZC3H4-dependent PTT, and therefore lack a requirement for SET1, because their transcription is driven by activation signals and other processivity factors which render the inhibitory effect of ZC3H4 on

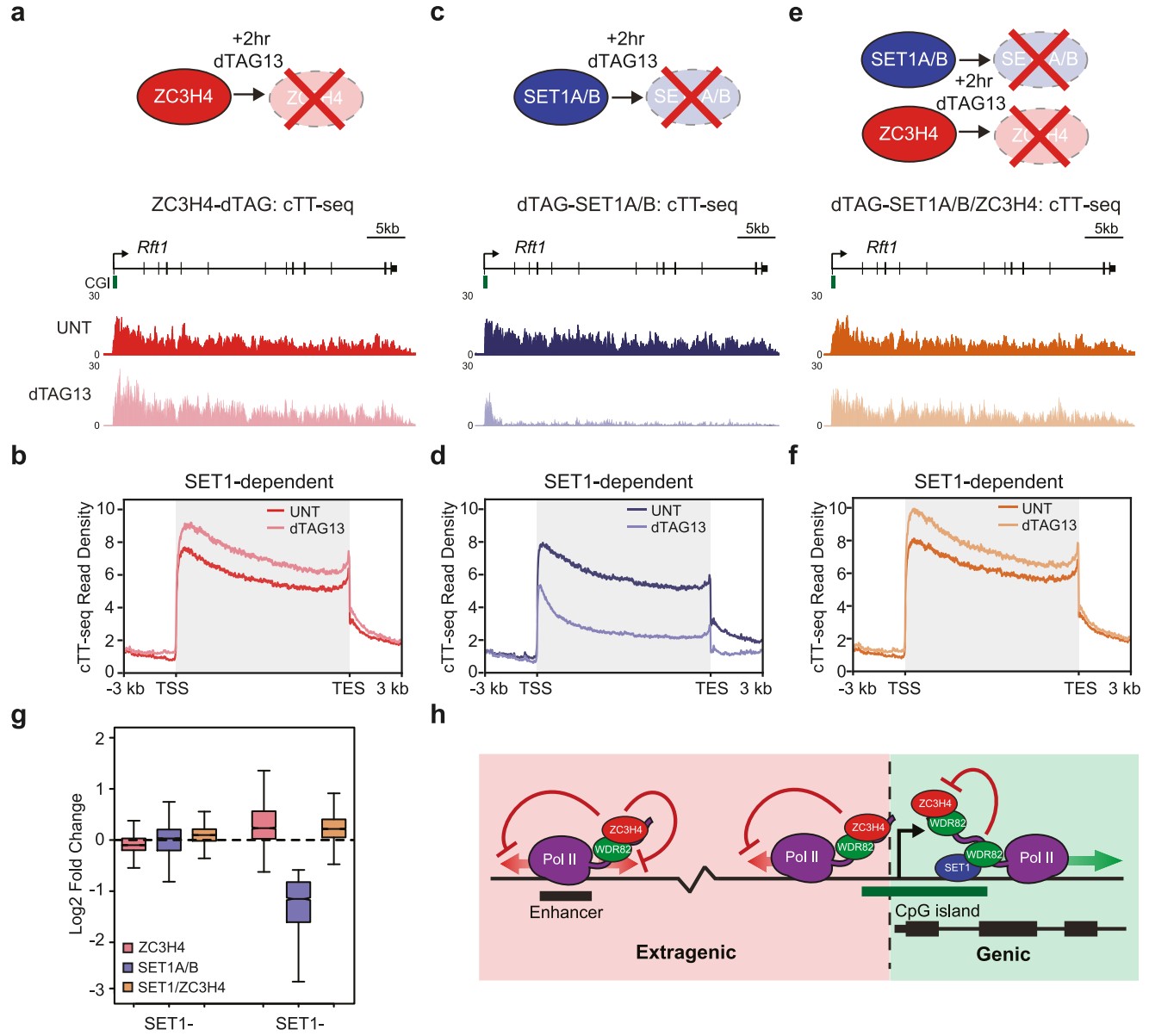

**Fig. 6 | SET1 complexes counteract premature transcription termination by ZC3H4. a** A schematic illustrating the ZC3H4-dTAG line (top panel) and a genomic snapshot showing cTT-seq signal at a SET1-dependent gene (*Rft1*) before (UNT, dark red) and after (light red) 2 h treatment with dTAG13 (bottom panels). **b** Metaplot analysis of transcription (cTT-seq) in the ZC3H4-dTAG line in cells that are either untreated (UNT) or treated with dTAG13 for 2 h at all transcribed SET1-dependent genes (*n* = 2633). **c**, **d** As per **a** and **b** but for the dTAG-SET1A/B line. **e**, **f** As per **a** and **b** but for the dTAG-SET1A/B/ZC3H4 line. **g** A box plot showing the log2 fold change in transcription at transcribed SET1-independent (*n* = 9151) and SET1-dependent (*n* = 2633) genes in the dTAG-SET1A/B, ZC3H4-dTAG, and dTAG-SET1A/B/ZC3H4 lines after treatment with dTAG13 for 2 h. The boxes show interquartile range, centre line represents median, whiskers extend by 1.5× IQR or the most extreme point (whichever is closer to the median), while notches extend by 1.58× IQR/sqrt(n), giving a

roughly 95% confidence interval for comparing medians. **h** A cartoon illustrating a model whereby WDR82-containing SET1 complexes bind to CpG-dense regions in CGIs downstream of TSSs to enable genic transcription by counteracting premature transcription termination by WDR82-containing ZC3H4 complexes. The defined mechanism through which SET1 complexes counteract the function of ZC3H4 complexes remains to be determined, but this likely involves both SET1 and ZC3H4 complexes interacting with the CTD of RNA Pol II and the integration of their distinct activities determining the effect on transcription. In contrast, extragenic transcription that emanates from regions lacking SET1 complex occupancy is subject to termination by WDR82-containing ZC3H4 complexes. In this model, CGIs and SET1 complex occupancy would distinguish genic from extragenic transcription, and protect genic transcription from premature transcription termination to enable gene expression.

transcription output nominal. Therefore, we propose that SET1 complex occupancy at CpG-dense regions downstream of actively transcribed TSSs distinguishes genic transcription from extragenic transcription, ensuring that low to moderately transcribed genes can be transcribed while non-productive extragenic transcription remains susceptible to termination by ZC3H4 (Fig. 6h).

In conclusion, we identify a gene regulatory mechanism whereby SET1 complexes bind to CpG-rich DNA in CGIs downstream of TSSs and counteract PTT by ZC3H4 complexes at low to moderately transcribed genes. Furthermore, we demonstrate that termination of early transcription by ZC3H4 complexes is widespread and show that pervasive PTT must be counteracted to support normal gene expression.

Importantly, this also reveals a role for CGIs and SET1 complexes in distinguishing genic from extragenic transcription and protecting genes from premature transcription termination to enable normal gene expression.

## Methods

### Cell culture

Mouse embryonic stem cells were grown in Dulbecco's Modified Eagle Medium (Thermo Fisher Scientific) supplemented with fetal bovine serum (FBS, 15% Biosera or 10% Sigma), 1× non-essential amino acids (Thermo Fisher Scientific), 2 mM L-glutamine (Thermo Fisher Scientific), 1× penicillin/streptomycin (Thermo Fisher Scientific), 0.5 mM beta-mercaptoethanol (Thermo Fisher Scientific), and 10 ng/ml leukaemia inhibitory factor (produced in-house). ESCs were grown on gelatinised plates at 37 °C and 5% $CO_2$. Cell lines expressing dTAG fusion proteins were treated with 100 nM dTAG-13 (produced by Behnam Nabet and Nathanael Gray[87] or Carole Bataille and Angela Russell) to induce protein depletion.

*Drosophila melanogaster* SG4 cells were grown adhesively at 25 °C in Schneider's *Drosophila* Medium (Thermo Fisher Scientific) supplemented with 1× penicillin/streptomycin and 10% heat-inactivated FBS (Biosera). Human HEK 293 T cells were grown at 37 °C and 5% $CO_2$ in Dulbecco's Modified Eagle Medium, supplemented with 10% FBS (Biosera), 1× penicillin/streptomycin, 2 mM L-glutamine and 0.5 mM beta-mercaptoethanol. All cell lines were routinely tested to ensure they were mycoplasma free.

### Stable cell line generation

To allow for their rapid depletion, we introduced an N-terminal dTAG into the endogenous *Set1a* and *Set1b* genes, and a C-terminal dTAG into the endogenous *Zc3h4* gene. To allow for efficient chromatin immunoprecipitation, we also introduced an N-terminal 3xT7-2xStrepII-dTAG tag into the endogenous *Set1a* gene and a C-terminal 2xStrepII-3xT7 tag into the endogenous *Zc3h4* gene. To generate the luciferase reporter cell line, we modified a previously described mouse ESC line containing a single copy insertion of a human gene desert bacterial artificial chromosome[104] with a cassette containing 7 CpG-free TetO DNA binding sites[104], followed by a CpG-free Ef1a promoter and a luciferase reporter gene[105].

Stable cell lines were engineered using CRISPR/Cas9-mediated genome editing. sgRNAs were designed using the CRISPOR online tool (http://crispor.tefor.net; Supplementary Table 1) and oligonucleotides encoding sgRNAs were cloned into the pSpCas9 (BB)-2A-Puro plasmid as previously described (Addgene #62988)[106]. Targeting constructs containing the sequence to be inserted and approximately 500 bp homology arms were cloned by Gibson Assembly (NEB). The FKBP12$^{F36V}$ tag (dTAG) was obtained from Addgene (#91797). ESCs were transfected at approximately 70% confluency in a 6-well plate with 0.5 μg of guide plasmid and 2 μg of targeting construct using Lipofectamine 3000, according to the manufacturer's protocol (Thermo Fisher Scientific). The following morning, transfected cells were passaged to new plates at low density and selected with 1 μg/ml puromycin for 48 h. Individual colonies were picked into 96-well plates and positive clones identified by PCR screens of genomic DNA.

### Luciferase reporter assays

To express rTetR-fusion proteins, fragments comprising GFP, full length SET1A and the SET1A catalytic mutant were Gibson assembled into a plasmid backbone containing a CAG promoter, FLAG-StrepII tag, nuclear localisation signal and rTetR, as described previously[104]. The rTetR was replaced with that from the TetON-3G plasmid for all SET1A minimal domain fusions and full length SET1A and SET1B NTD in Fig. 3 (Addgene plasmid #96963)[107,108].

ESCs were transfected at approximately 70% confluency with Lipofectamine 2000, according to the manufacturer's protocol (Thermo Fisher Scientific). For preparation of nuclear extract, cells were transfected in 6-well plates with 2.5 μg plasmid. For luciferase assays, cells were transfected in 96-well plates with 98 ng rTetR plasmid and 2 ng pRL *Renilla* Luciferase control reporter plasmid (Promega) to control for transfection efficiency. Each transfection was performed in six wells of a 96-well plate to obtain 3 technical replicates for both untreated and treated with doxycycline. Following overnight transfection of plasmids expressing rTetR-fusion proteins and *Renilla* luciferase, three wells for each transfection were treated with 1 μg/ml doxycycline for 6 h. Luciferase reporter assays were performed with the Dual-Luciferase Reporter Assay System (Promega). In brief, cells were lysed in 20 μl 1× passive lysis buffer by shaking at room temperature for 20 min. Totally, 10 μl cell lysate was added to 50 μl Luciferase Assay Reagent II and Firefly luciferase measured using a 10 s measurement in a Luminometer. Totally, 50 μl Stop & Glo Reagent was added and *Renilla* luciferase measured using a 10 s measurement. Each Firefly reading was normalised to the respective *Renilla* reading. Technical replicates were averaged and normalised to the readings obtained in the absence of doxycycline. Each assay was performed in at least biological triplicate.

### Protein extraction and immunoblotting

To prepare histone extracts, pelleted cells were washed in RSB (10 mM Tris HCl pH 7.4, 10 mM NaCl, 3 mM $MgCl_2$), centrifuged at 240 × *g* for 5 min and resuspended in RSB supplemented with 0.5% NP-40. Following incubation on ice for 10 min, cells were centrifuged at 500 × *g* for 5 minutes. The nuclear pellet was resuspended in 5 mM $MgCl_2$, an equal volume of 0.8 M HCl added, and then incubated on ice for 20 min to extract histones. After centrifugation for 20 min at 18,000 × *g*, the supernatant was taken and histones precipitated by adding TCA to 25% v/v and incubating on ice for 30 min. Histones were pelleted by centrifugation at 18,000 × *g* for 15 min, and the pellet was washed twice in cold acetone. The histone pellet was resuspended by gentle vortexing in 1× SDS loading buffer (2% SDS, 100 mM Tris pH 6.8, 100 mM DTT, 10% glycerol, 0.1% bromophenol blue) and boiling at 95 °C for 5 min. Any insoluble precipitate was pelleted by centrifugation at 18,000 × *g* for 15 min and the soluble fraction taken as the histone extract. Histone extract concentrations were compared across samples by SDS-polyacrylamide gel electrophoresis (SDS-PAGE) and Coomassie Blue staining.

To prepare nuclear extracts, cell pellets were resuspended in 10× pellet volumes of buffer A (10 mM HEPES pH 7.9, 1.5 mM $MgCl_2$, 10 mM KCl, 0.5 mM DTT, 0.5 mM PMSF, 1x cOmplete protease inhibitor cocktail (PIC, Roche)) and incubated for 10 min on ice. After centrifugation at 500 × *g* for 5 min, the cell pellet was resuspended in 3× pellet volumes of buffer A supplemented with 0.1% NP-40 and incubated on ice for 10 min. Nuclei were pelleted at 1500 × *g* for 5 min and then resuspended in 1× pellet volume of buffer C (250 mM NaCl, 5 mM HEPES pH 7.9, 26% glycerol, 1.5 mM $MgCl_2$, 0.2 mM EDTA, 0.5 mM DTT, 1× PIC). The volume of the nuclear suspension was measured and the NaCl concentration increased to 400 mM by dropwise addition of 5 M NaCl. Nuclei were incubated at 4 °C for 1 h with gentle inversion to extract nuclear proteins. After centrifugation at 18,000 × *g* for 20 min, nuclear proteins were recovered in the supernatant. Protein concentration was determined by Bradford assay (BioRad) and typically 25 μg was used for western blotting.

Protein extracts were resolved using either home-made SDS-PAGE gels or 3–8% NuPAGE Tris-Acetate gels (Thermo Fisher Scientific), when analysing proteins of a molecular weight >180 kDa. Typically, proteins were transferred to nitrocellulose membranes by semi-dry transfer using the Trans-Blot Turbo Transfer System (BioRad). Transfer was performed as per the manufacturer's guidelines, depending on the size of the proteins being transferred. Membranes were typically imaged using an Odyssey Fc system (LI-COR) and images were analysed using Image Studio v5.2. Changes in bulk protein levels were

quantified relative to those of loading controls. For SET1B western blots, proteins were transferred to 0.45 μm nitrocellulose membranes by wet transfer. Transfer was performed in 1x wet transfer buffer (25 mM Tris, 192 mM glycine, 10% Methanol, 1% SDS) at 100 V for 90 min at 4 °C. Membranes were developed by chemiluminescence.

Antibodies used for western blot analysis are in Supplementary Table 2[109].

## Co-Immunoprecipitation

Immunoprecipitations (IPs) of SET1A were performed using 550 μg of nuclear extract. Extracts were diluted to 500 μl with BC150 (150 mM KCl, 10% glycerol, 50 mM HEPES pH 7.9, 0.5 mM EDTA, 0.5 mM DTT and 1× PIC), with 250 units Benzonase Nuclease (Sigma). Protein A beads (Repligen) were blocked in BC150 supplemented with 1% Fish gelatin (Sigma) and 0.2 mg/ml BSA (NEB) for 1 h at 4 °C. Extracts were pre-cleared with 50 μl slurry blocked beads for 1 h at 4 °C and then incubated rotating with 20 μl SET1A antibody (Klose Lab) or 5 μl FLAG antibody (Sigma F1804) overnight at 4 °C. Totally, 50 μl slurry blocked Protein A beads were used to precipitate antibody-bound protein at 4 °C for 3 h. Beads were pelleted at $1000 \times g$, and washed 3 times with BC150 + 0.02% NP-40, with one final wash in BC150. To elute the immunoprecipitated complexes, beads were resuspended in 2× SDS loading buffer, boiled at 95 °C for 5 min, and the supernatant collected for western blotting. An appropriate amount of nuclear extract was taken as the input sample and inputs were incubated in 1× SDS loading buffer at 95 °C for 5 min. When probing for interacting proteins of interest smaller than 50 kDa, HRP-conjugated VeriBlot secondary antibodies (Abcam) were used to avoid cross-reactivity with denatured IgG. Membranes were then imaged by chemiluminescence.

To examine the SET1A DPR/AAA mutation, HEK 293 T cells were transfected with plasmids expressing WT/mutated SET1A NTD constructs using Lipofectamine 2000 according to the manufacturer's instructions. Cells were passaged the following day and allowed to grow for a further 24 h before harvesting by trypsinisation. Totally, 600 μg nuclear extract was used as input for each IP. Extracts were diluted in nuclear extraction buffer C without salt to give a final NaCl concentration of 150 mM. Benzonase nuclease (125U) was added and extracts were incubated for 30 min at 4 °C with gentle mixing. Samples were centrifuged at $21,000 \times g$ for 5 min and supernatant was used as input for IPs. Totally, 25 μl anti-FLAG M2 affinity resin (Sigma A2220) was used for each IP. Beads were washed three times in BC150 then incubated with extracts for 4 h at 4 °C with gentle agitation. The beads were then washed 3 times in BC150 with 0.02% NP40 and bound proteins were eluted for western blotting by boiling for 5 min in 35 μl 2× SDS-PAGE loading dye. An HRP-conjugated anti-FLAG antibody (Sigma, A8592) was used to probe for the SET1A NTD fragments and the membranes were imaged by chemiluminescence.

## Size exclusion chromatography

Nuclear extract was treated with Benzonase (250U Benzonase per mg nuclear extract) and dialysed overnight into BC200 buffer (50 mM HEPES pH 7.9, 200 mM KCl, 10% Glycerol, 1 mM DTT). 2 mg dialysed nuclear extract was loaded on a Superose 6 Increase 10/300 GL column (Cytiva, precalibrated with dextran blue, Mix 1 (Ferritin, 440 kDa; Conalbumin, 75 kDa), and Mix 2 (Thyroglobulin, 669 kDa; Aldolase, 158 kDa; Ovalbumin, 43 kDa)) and run in BC200 buffer at 0.2 ml/min. Eluate was collected in 250 μl fractions. Protein fractions were precipitated with trichloroacetic acid and 15% of the fraction was loaded onto an SDS-PAGE gel for analysis by western blot.

## Calibrated Total RNA-sequencing (cRNA-seq)

ESCs (~$10^6$) were counted and mixed with ¼ of the number of SG4 Drosophila cells in PBS. RNA was extracted from cells using TRIzol reagent, according to the manufacturer's protocol (Thermo Fisher Scientific). gDNA contamination was depleted using the TURBO

DNA-free Kit (Thermo Fisher Scientific) and the quality of RNA was assessed using a 2100 Bioanalyzer RNA 6000 Pico kit (Agilent). 900 ng RNA was depleted of rRNA using the NEBNext rRNA Depletion kit (NEB). RNA-seq libraries were prepared from an equal amount of ribo-depleted RNA using the NEBNext Ultra II Directional RNA Library Prep kit, including 2.5–4 min fragmentation at 94 °C (NEB).

## Calibrated transient transcriptome-sequencing (cTT-seq)

cTT-seq was performed largely as described previously[99]. In brief, 9 million ESCs and 3 million Drosophila SG4 cells were labelled with 500 μM 4-thiouridine (4sU, Glentham Life Sciences) for 15 min and harvested into TRIzol reagent. 4sU-labelled mouse and Drosophila cells were mixed and RNA was extracted using the Direct-zol DNA/RNA Miniprep kit (Zymo Research) as per the manufacturer's protocol. gDNA was depleted using the TURBO DNA-free Kit (Thermo Fisher Scientific). An equal quantity of RNA (60–80 μg) was taken into 100 μl nuclease free water and fragmented on ice with 20 μl 1 M NaOH for 20 min. Fragmentation was stopped with 80 μl 1 M Tris, pH 6.8 and the RNA was cleaned up with Micro Bio-Spin P-30 gel columns (Biorad). RNA was biotin-labelled with 50 μl 0.1 mg/ml MTSEA biotin-XX linker (Biotium) with 3 μl biotin buffer (833 mM Tris HCl, pH 7.4, 83.3 mM EDTA) for 30 min at RT. Biotin-labelled RNA was purified with a 1:1 ratio of phenol/chloroform/isoamyl alcohol (Thermo Fisher Scientific). Streptavidin pull-down was performed with the μMACS Streptavidin Kit (Miltenyi Biotec), washing the columns three times with 55 °C pull-down wash buffer (100 mM Tris HCl, pH 7.4, 10 mM EDTA, 1 M NaCl and 0.1% Tween 20) and 3x RT pull down wash buffer. Biotin-labelled RNA was eluted with 100 μl elution buffer (100 mM DTT in nuclease-free water) and cleaned up with the RNeasy MinElute Cleanup kit (QIAGEN), adjusting the amount of ethanol to capture RNA < 200 nucleotides in length. RNA was quantified using the Qubit RNA HS assay kit and RNA libraries were prepared from 20-50 ng RNA with the Ultra II Directional RNA library prep kit, as per the manufacturer's guidelines for rRNA depleted and FFPE RNA (NEB).

## Native cChIP-sequencing

Native cChIP-seq was performed as described previously[110]. In brief, $5 \times 10^7$ ESCs were mixed with $2 \times 10^7$ Drosophila SG4 cells and nuclei were released by resuspending in RSB (10 mM Tris HCl pH 8, 10 mM NaCl, 3 mM MgCl$_2$) with 0.1% NP40. Nuclei were pelleted at $1500 \times g$ for 5 min and then washed and resuspended in 1 ml MNase digestion buffer (RSB with 0.25 M Sucrose, 3 mM CaCl$_2$, 1x PIC). Each sample was incubated with 200 units of MNase (Fermentas) at 37 °C for 5 min, with gentle inversion. Digestion was stopped by addition of 4 mM EDTA. Following centrifugation at $1500 \times g$ for 5 min, the supernatant (S1 fraction) was retained and the remaining pellet was resuspended in 300 μl nucleosome release buffer (10 mM Tris HCl pH 7.5, 10 mM NaCl, 0.2 mM EDTA, 1× PIC), rotated at 4 °C for 1 h and then passed five times through a 27 G needle using a 1 ml syringe. Following centrifugation at $1500 \times g$ for 5 min, the supernatant (S2) was combined with S1 fraction, aliquoted, snap frozen and stored at −80 °C. Digestion to predominantly mononucleosomal fragments was confirmed by agarose gel electrophoresis of purified DNA.

For each IP, 100 μl S1/S2 nucleosomes were diluted to 1 ml total volume in native ChIP incubation buffer (70 mM NaCl, 10 mM Tris HCl, pH 7.5, 2 mM MgCl$_2$, 2 mM EDTA, 0.1 % Triton X-100, 1x PIC) and immunoprecipitated with 1.5 μl H3K4me3 antibody (Klose Lab) overnight at 4 °C. IPs were all set up in duplicate for each sample. 100 μl diluted chromatin was also set aside as an input sample. Protein A agarose beads (Repligen) were blocked with 1 mg/ml BSA and 1 mg/ml yeast tRNA in native ChIP incubation buffer, overnight at 4 °C. 40 μl slurry of pre-blocked agarose beads were used to capture antibody-bound nucleosomes at 4 °C for 1 h. Beads were then washed 4× with Native ChIP wash buffer (20 mM Tris HCl, pH 7.5, 2 mM EDTA, 125 mM NaCl, 0.1 % Triton X-100, 1x PIC) and 1x with TE buffer, pH 8. DNA was

eluted by vortexing for 30 min in elution buffer (1% SDS and 0.1 M NaHCO₃) and DNA was purified using a ChIP DNA Clean & Concentrator kit (Zymo Research). For each ChIP, DNA from the matched input control (10% of the IP) was also purified. Purified DNA was analysed using ChIP-qPCR and cChIP-seq libraries for both ChIP and input samples were prepared using NEBNext Ultra II DNA Library Prep Kit for Illumina following the manufacturer's guidelines (NEB).

### Cross-linked cChIP-sequencing

For double cross-linked T7-SET1A ChIP and ZC3H4-T7 ChIP, $5 \times 10^7$ ESCs were fixed with 2 mM DSG (disuccinimidyl glutarate, Thermo Fisher Scientific) for 50 min at 25 °C and then 1% formaldehyde (methanol-free, Thermo Fisher Scientific) for 10 min. Alternatively, for single cross-linked RNA Pol II ChIP, $5 \times 10^7$ ESCs were fixed with 1% formaldehyde for 10 min at 25 °C. Fixation was quenched using glycine added to 125 mM. Cells were then pelleted at 1000 x g for 5 min and washed with PBS. Cross-linked ESCs were mixed with $1 \times 10^5$ cross-linked HEK 293 T/T7-SCC1 cells (1% formaldehyde, 15 min for SET1A ChIP; a gift from Martin Houlard, Nasmyth lab) or $2 \times 10^6$ cross-linked HEK 293 T cells (1% formaldehyde, 10 min for RNA Pol II ChIP). Chromatin was prepared by incubation in 1 ml FA-lysis buffer (50 mM HEPES pH 7.9, 150 mM NaCl, 1 mM EDTA, 0.5 mM EGTA, 0.5% NP-40, 0.1% Na-deoxycholate, 0.1% SDS, 1x PIC, 1 mM AEBSF. For RNA Pol II ChIP, EDTA concentration was increased to 2 mM and 10 mM NaF was added fresh) on ice for 10 min. Chromatin was sonicated using a Bioruptor Pico sonicator (Diagenode) at 4 °C. Sonication was performed using 23–30 cycles of 30 s on/30 s off at full power, shearing genomic DNA to an average size of 0.5 kb. The sonicated material was pelleted at 20,000 × g for 20 min, and the supernatant taken as sonicated chromatin.

Totally, 300 µg chromatin was used per IP. Chromatin was diluted to 1 ml total volume per IP in FA-lysis buffer. An additional volume of diluted chromatin was taken to use as an input sample. Protein A agarose beads (Repligen) were blocked with 1 mg/ml BSA and 1 mg/ml yeast tRNA in 1× TE buffer at 4 °C for 1 h. Chromatin was pre-cleared with agarose beads (40 µl slurry beads per ChIP) at 4 °C for 1–2 h. The input sample was taken from the pre-cleared chromatin, and the remainder was immunoprecipitated overnight at 4 °C with the appropriate amount of antibody: T7 (Cell Signalling, D9E1×, 10 µl) or RNA Pol II N-terminal domain (Cell Signalling, D8L4Y, 15 µl). Antibody-bound chromatin was isolated for 3 h at 4 °C using 40 µl slurry of blocked Protein A agarose beads. Washes were carried out for 5 min each at 4 °C, using FA-lysis buffer, FA-lysis buffer with 500 mM NaCl, 1× DOC buffer (10 mM Tris HCl, pH 8, 250 mM LiCl, 1 mM EDTA (2 mM EDTA for RNA Pol II ChIP), 0.5% NP-40, 0.5% Na-deoxycholate), and 2 washes with TE buffer (1× PIC and 1 mM AEBSF were added fresh to all wash buffers. 10 mM NaF was also added for RNA Pol II ChIP). DNA was eluted by vortexing for 30 min in elution buffer (1% SDS and 0.1 M NaHCO₃). Cross-links were reversed for ChIPs and inputs at 65 °C overnight with 200 mM NaCl and 2 µl RNase A (Sigma). Samples were then incubated with 20 µg Proteinase K for 1 h at 45 °C. DNA for ChIPs and inputs was purified using a ChIP DNA Clean & Concentrator kit (Zymo Research). Purified DNA was analysed using ChIP-qPCR. cChIP-seq libraries for both ChIP and input samples were prepared using NEBNext Ultra II DNA Library Prep Kit for Illumina following manufacturer's guidelines (NEB).

### Massively parallel sequencing

All sequencing experiments were carried out in at least biological triplicate. Sequencing samples were indexed using NEBNext Multiplex Oligos (NEB). The average size and concentration of all sequencing libraries was analysed using a Bioanalyser High Sensitivity DNA Kit (Agilent) followed by qPCR using SensiMix SYBR Green mastermix (Bioline) and KAPA Illumina DNA standards (Roche). Libraries were sequenced using a NextSeq500 (Illumina). ChIP and TT-seq libraries were sequenced with 40 bp paired-end reads and RNA-sequencing libraries with 80 bp paired-end reads.

### Read alignment and normalisation

For cChIP-seq, paired-end reads were aligned to the concatenated mouse mm10 and spike-in genomes (mm10 + dm6 for Native cChIP and mm10 + hg19 for cross-linked cChIP) using Bowtie2 with the '-no-mixed' and '-no-discordant' options[111]. Reads that mapped more than once were discarded and PCR duplicates were removed using Sambamba[112]. For cRNA-seq and cTT-seq, reads that aligned to the mm10 and dm6 rDNA genomic sequences (GenBank: BK000964.3 and M21017.1) were first identified using Bowtie2 with '-very-fast', '-no-mixed' and '-no-discordant' options) and discarded[111]. Unmapped reads were then aligned to the concatenated mm10 and dm6 genomes using STAR[113]. To improve mapping of intronic sequences, reads that failed to map using STAR were aligned using Bowtie2 with '-sensitive-local', '-no-mixed' and '-no-discordant' options. Uniquely aligned reads from the last two steps were combined for further analysis and PCR duplicates were removed using Sambamba (Supplementary Data 1).

Sequencing datasets were calibrated to the spike-in *Drosophila* or human genomes, as described previously[110,114–116]. For cChIP-seq, the number of mm10 reads were randomly downsampled to reflect the total number of dm6 or hg19 reads in that sample. Furthermore, to adjust for any variation in cell mixing, each sample was adjusted using the ratio of spike-in reads relative to mm10 reads in the relevant input sample. ZC3H4 ChIP-seq was performed without spike-in normalisation. For cRNA-seq and cTT-seq, the number of mm10 reads were randomly downsampled to reflect the total number of dm6 reads in that sample. After normalisation, read coverages for individual biological replicates were compared across regions of interest using the multiBamSummary and plotCorrelation functions from deepTools (version 3.1.1)[117]. Biological replicates correlated well with each other (Pearson correlation coefficient > 0.9) and were merged for subsequent analysis. Genome coverage tracks were generated using the pileup function from MACS2 for cChIP-seq and genomeCoverageBed from BEDTools (version 2.17.0) for cRNA-seq and visualised using the UCSC genome browser[118–122]. Differential genome coverage tracks (fold change or log2 fold change of two conditions) were obtained using the bigwigCompare function from deepTools[117].

### Peak calling and annotation

H3K4me3 peak sets were generated from each ChIP replicate using MACS2[119,122] ('BAMPE' and 'broad' options specified), with a matched input sample from each biological replicate used for background normalisation. A H3K4me3 peak set comprising an overlap of the peaks called in three biological replicates in untreated samples in dTAG-SET1A/B cells were used for further analysis. TSS-associated H3K4me3 peaks were identified using BEDTools intersect[120], only carrying forward peaks which overlapped directly with a TSS from a custom-built, non-redundant mm10 set[116] ($n = 20,633$; TSS-associated H3K4me3 peaks $n = 14,065$). To annotate genes as CGI-associated, non-methylated islands were identified using MACS2 peak calling from BioCAP-seq data using a matched input control ($n = 27,047$)[123]. BEDTools Intersect was used to identify genes with a TSS which is within 1.5 kb of a non-methylated island ($n = 14,438$) and non-methylated islands which are within 1.5 kb of a TSS ($n = 15,259$). Enhancer regions were previously defined from H3K27ac cChIPseq and ATAC-seq[124].

### Read count quantification and analysis

Differential gene expression analysis was performed as described previously[88]. In brief, mm10 read counts were obtained from individual biological replicates prior to spike-in normalisation using a SAMtools-based custom Perl script within the non-redundant mm10 gene set ($n = 20,633$), 5 kb regions of TSS upstream antisense transcription or 10 kb regions around enhancers ($n = 4156$). dm6 read counts were obtained from a set of unique dm6 refGene genes and were used to calculate normalisation size factors using the DESeq2 package[125]. These size factors were then applied to DESeq2 analysis of the mm10 read

counts. A change was considered significant based on a threshold of p-adj < 0.05 and fold change > 1.5. For visualization purposes, DESeq2-normalized read counts were averaged across the replicates and used to calculate RPKM (Reads per kilobase per million).

For H3K4me3 cChIP-seq, mm10 read counts from individual biological replicates prior to spike-in normalisation within H3K4me3 peaks were obtained using BEDTools multicov[120]. Spike-in reads were pre-normalised using the dm6/mm10 ratio in the corresponding input sample and reads counted in dm6 TSS ± 2 kb of a custom unique set. Differential enrichment analysis was performed using DESeq2, as described above for cRNA-seq[125].

Metaplot and heatmap analysis was performed using computeMatrix and plotProfile/plotHeatmap from deepTools[117]. Metaplot profiles represent the mean read density over a set of genomic regions. To ensure strandedness for cTT-seq, computeMatrix was used to obtain sense and antisense read density within required regions on each strand. Stranded matrices were then merged using computeMatrixOperations.

Boxplots and scatterplots were made using R and ggplot2. The boxes for boxplots show interquartile range, centre line represents median, whiskers extend by 1.5× IQR or the most extreme point (whichever is closer to the median), while notches extend by $1.58 \times IQR/sqrt(n)$, giving a roughly 95% confidence interval for comparing medians. Pearson and spearman correlation coefficients were computed using the 'cor.test' function, scatterplots were coloured by density with 'stat_density2d' and the linear regression line was plotted using 'geom_smooth'. Student's $t$ tests were also performed in R with samples considered to be paired and two-sided alternative hypothesis, unless otherwise stated. Gene Ontology analysis was performed in R using the 'enrichGO' function from clusterProfiler[126] and with the complete non-redundant mm10 gene set as background.

### Multiple sequence alignments
Multiple sequence alignments of protein sequences were generated using MUSCLE[127] with default settings. Alignments were then visualised using Jalview[128].

### RNA in situ hybridization protocol and imaging
Probes targeting exonic sequences were designed with Stellaris probe designer (Stellaris) such that they cover evenly all exons of a gene studied. Totally, 40–48 oligos were labelled by terminal deoxynucleotidyl transferase using dideoxy-UTP conjugated with Atto565[129]. Cells were trypsinised prior to fixation with 3.7% formaldehyde for 15 min, followed by incubation in 70% ethanol for 1 h. RNA-FISH labelling was performed in suspension in the presence of 2× SSC, 10% formamide and 20% dextran sulfate at 37 °C overnight. The cells were then centrifuged, washed 3× to diminish non-specific signal, and fluorescently labelled with saturating DAPI and Agglutinin-Alexa488 concentrations. The cell suspension was mixed 1:1 with Vectashield H-1000 (Vectorlabs) and introduced between a glass slide and #1.5 coverslip. Three-dimensional 3 colour images were acquired using the Olympus IX83 system and cellSens software using 63× objective lens with extra 2× magnifying lens resulting in 91.5 nm camera pixel size. Images were exported, single cells were automatically thresholded based on DAPI and Agglutinin, and transcripts were identified in 2D maximal projections of 3D RNA-FISH images using publically available software ThunderFISH: https://github.com/aleks-szczure/ThunderFISH. This procedure has proven effective in acquiring more than 400 cells per sample, per biological replicate (total $n = 3$ biological replicates were acquired for each gene). The detailed protocol of RNA-FISH and characterization of the analysis method have been previously described in detail[88].

### Reporting summary
Further information on research design is available in the Nature Portfolio Reporting Summary linked to this article.

## Data availability
The data that support this study are available from the corresponding author upon reasonable request. High-throughput sequencing datasets generated for this study are available in the GEO database under the accession number (GSE199805). Published data used in this study include BioCAP-seq data (GSE43512) from[123], enhancer annotations (GSE161996) from[124], CpG density tracks from[130], SET1A ChIP-seq (GSE98140) data from[131] and mRNA half-life data (GSE86336) from[132]. Source data are provided with this paper.

## Code availability
Scripts for aligning and normalising high-throughput sequencing data are available at https://github.com/nFursova/Calibrated_ChIPseq_RNAseq. A custom-made ImageJ script for preprocessing 3D images (ThunderFISH) is publicly available with a detailed manual for sample preparation and script use at https://github.com/aleks-szczure/ThunderFISH.

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

## Acknowledgements
We would like to thank Tom Milne, Krzysztof Kus, Lars Jansen, and Alan Au for critical reading of the manuscript, as well as members of the Klose lab for fruitful discussions. We thank Angelika Feldmann, Nadezda Fursova, and Paula Dobrinic for helpful discussions regarding computational analysis. We are grateful to Amanda Williams at the Department of Zoology, Oxford, for sequencing support on the NextSeq 500. We would like to thank Behnam Nabet, Nathanael Gray, Carole Bataille and Angela Russell for generous provision of dTAG13 compound. We thank Martin Houlard for providing the T7-SCC1 cell line and Michael Rehli for providing the pCpGL-CMV/EF1 plasmid. Work in the Klose lab is supported by the Wellcome Trust (209400/Z/17/Z). A.L.H. (203829/Z/16/A) and A.H.T. (102349/Z/13/Z) were supported by Wellcome Trust studentships. J.R.K. is supported by the Oxford-Wolfson Marriott Graduate Scholarship.

## Author contributions
A.L.H. and R.J.K. conceived the project and wrote the paper with contributions from all co-authors. A.L.H. performed most of the experiments, data analysis, and visualisation. A.T.S. performed the smRNA-FISH, data analysis, and visualisation. J.R.K. performed the WDR82 interaction analysis, multiple sequence analysis, and visualisation. A.H.T. generated the gene expression reporter cell line. A.L., E.D., and N.P.B. contributed to cell line generation, experimentation, and drafting the paper. N.P.B. contributed to visualisation of data in figures. R.J.K. supervised the project.

## Competing interests
The authors declare no competing interests.
