## [Peer Review File · Nature Communications]

REVIEWER COMMENTS

Reviewer #1 (Remarks to the Author):

In recent years, premature termination of transcription (PTT) has emerged as a prevalent process in human cells. In this paper, Hughes et al show that SET1 protects thousands of genes from PTT via the recently discovered termination factor, ZC3H4. Using the dTAG degron, they start by showing that 2000-odd genes show reduced mRNA levels after the rapid loss of SET1A. Fewer are affected by SET1B loss whereas co-depletion of SET1A and B affects the most genes. Basically, it looks like SET1A is more important than SET1B for this effect. The transcripts affected have CGI promoters and tend to be lowly expressed. Despite SET1A being widely studied as a histone methyltransferase, H3K4 methylation was essentially unaffected by its elimination. Furthermore, its promotion of gene transcription, in a tethered function assay, is unimpacted by mutation of its methyltransferase domain. Instead, it is proposed that SET1A promotes transcription by interaction with WDR82. This prevents the termination factor, ZC3H4, in complex with WDR82, from causing PTT at these genes. Consistently, transcriptional downregulation of SET1A-affected genes is reversed by co-depletion of ZC3H4. Overall, I found this to be an interesting and well-written paper - suitable for publication should the authors address the concerns/comments below.

1. Fig 1 shows that 2000-odd transcripts show reduced levels after rapid (2hr) SET depletion. This experiment is described as total RNA seq. If so, doesn't this mean that the SET1-affected transcripts must have very short half lives (for a steady state reduction to be seen after a relatively short period of time)? This seems like an interesting observation worthy of discussion unless I misunderstand it.
2. The lack of effect of SET1 loss on H3K4 is very interesting given the long-held view that it is a methyltransferase. The TetO tethering experiment is also a very useful way of demonstrating that this activity is unlikely to underpin the effect on transcription. This lack of effect on methylation (and of this on transcription) seems important and goes against the grain. As such, I think an additional experiment is warranted whereby mutated SET1 is used to rescue the effects of SET1A depletion on transcription. This need not be a full sequencing experiment since the authors have already identified thousands of SET1 transcription targets, a sample of which could be used for this experiment.
3. Related to the above (minor point), could other H3K4 transferase activities substitute the loss of SET1A/B even after rapid loss?
4. Figure 3 clearly shows that just the WDR82 binding part (NTD) of SET1A is sufficient for its transcriptional effects (to drive luciferase expression). Presumably, this means the effect in cells could be due to some other (WDR82-interacting?) factor rather than SET1A that is pulled in by WDR82-SET1A. I say this because the DPR fragment that cannot bind WDR82 is unable to drive luciferase expression. Can this be discussed/speculated on or do the authors have an idea of what it might be?
5. ZC3H4 depletion also causes upregulation of many sense transcripts. What is the overlap between this set and those affected by SET1A loss? Would it be predicted that they would not overlap given the proposed role of SET1A in suppressing ZC3H4 effects? FigS6 has some of this information but additional information for the SET1-independent genes would be more complete and may add to the data shown for ZC3H4.
6. Related to 5, are those genes affected by ZC3H4 less likely to have strong CGIs (i.e. the opposite relationship to the SET1 affected)?
7. ChIP-seq shows ZC3H4 presence at promoters of SET1-dependent and independent genes whereas the model in Fig 6 depicts that SET1 opposes the recruitment of ZC3H4 to the promoters of SET1-dependent genes. In figure 5C there is more ZC3H4 recruited to the coding direction of the SET1 dependent genes vs the independent set. Isn't this inconsistent with the model as drawn? This suggests that ZC3H4 is recruited to promoters regardless of whether they are susceptible to SET1 effects, which isn't what the model implies.

8. In yeast, WDR82 links SET1 to Pol II. Could this be important in the present paper as an additional/alternative mode of recruitment vs CGIs?

Reviewer #2 (Remarks to the Author):

The authors present a manuscript that attempts to delineate the mode of action of SET1 complexes in transcription regulation. Throughout the study, rapid depletions are used to focus on direct targets and limit pleiotropic effects. This allows the authors to separate the effects of H3K4me3 loss and loss of SET1 proteins itself. Surprisingly, this reveals an important, potentially methyltransferase-independent role of SET1 in regulation of lowly-expressed genes. The redundancy of methyltransferase activity for this regulation is elegantly confirmed by a series of in vivo reporter assays, but not at the genome-wide level as the rest of the study. The authors show data supporting the view that SET1 antagonizes ZC3H4-dependent premature transcription termination. This is supported by examining the effects of ZC3H4 depletion, and simultaneous depletion of ZC3H4 and SET1. The work is original, overall well executed, clearly presented, and would be of interest to several fields of research in molecular biology and biomedicine. Nevertheless, the manuscript would greatly benefit from some additional experiments, analyses, clarifications, and additional mechanistic examination. Please see the point-by-point comments for details.

Major points:

1. "The addition of the dTAG did not affect SET1A protein levels (Fig.1a), SET1A complex formation (Supplementary Fig.1a), or its preferential localisation to CpG-rich regions downstream of TSSs at expressed CGI-associated gene promoters"

To properly support the claim that tagging SET1A with FKBP12 doesn't affect preferential localisation, it would be necessary to show a side-by-side ChIP-seq analysis of tagged and untagged line.

The authors claim that SET1A complex formation is not affected, but IP efficiencies in FigS1A appear slightly reduced, including the bait protein. Could the authors provide a more even blot?

2. "Interestingly, changes in gene expression were less pronounced at later time points after SET1A depletion, suggesting that additional mechanisms may compensate for its depletion over time (Supplementary Fig.1e)."

For clarity, authors should specify what "additional mechanisms" refers to. For instance, do they suggest the effect is seen due to technical reasons, such as selection for cells with poorer depletion in cell culture, or some other cell-intrinsic biological phenomenon, or both.

3. Figure 1k and Figure S1h+j. It is clear that SET1A/B depletion predominantly results in gene downregulation, but the authors should include the upregulated gene category in these analyses, particularly because this category is included in Figure 2c.

4. Figure S3A – The NTD of SET1B appears to have an effect several fold higher than the NTD of SET1A, at similar expression levels. The authors make no comment on this, and currently the phrasing in the text implies the effects to be equivalent. One potential explanation for this is that SET1B-NTD (and therefore probably the protein itself) is a stronger WDR82 interactor. The manuscript would benefit from at least some test whether

this is the case.

5. A major conclusion of this study is that SET1 functions independently of its methyltransferase activity, which is supported by the SET1A tethering reporter assay. To indeed generalise such conclusion, it is imperative to show this genome-wide. Therefore, the authors must demonstrate that the SET1A-depletion phenotype can be rescued by ectopic expression of the inactive form of the protein.

6. Same critique as raised in point 5 applies to Figure 3. Here it must be demonstrated genome-wide that ectopic expression of the DPR/AAA mutant does not rescue SET1A depletion.

7. The findings described in Figure 6 and associated text suggest that loss of ZC3H4 is dominant in relation to the loss of SET1 when it comes to transcriptional output of SET1-dependent genes. The authors conclude that therefore the primary function of SET1 in this context is to antagonize ZC3H4. However, degradation of ZC3H4 on its own results in a minor increase in transcription. A possible explanation for that is that WDR82 is destabilised upon ZC3H4 depletion. In this case, potential direct activity of SET1 promoting transcription elongation would be lost, possibly partially compensating for the loss of ZC3H4. The authors should address this possibility by including a western blot for WDR82 under the same conditions as in FigS6B. The authors should also show whether the chromatin binding of ZC3H4 is dependent on SET1 and vice versa.

8. In the discussion, the authors propose that SET1-WDR82 can antagonize the activity of ZC3H4-WDR82. While the functional antagonism is described, it is not made clear how it is achieved mechanistically. The manuscript would benefit greatly from further mechanistic investigation. The simplest mechanism, which is also hinted in the model in Fig6h would be that SET1 and ZC3H4 compete for WDR82 binding. One way this could be tested is by a series of co-IP experiments, wherein SET1 or ZC3H4 are depleted, and the other factor immunoprecipitated. If there is competition, the amount of WDR82 pulled down should increase in both conditions compared to non-depleted. Otherwise, overexpression of the NTD of SET1 might cause reduced pulldown efficiencies for both endogenous SET1 and ZC3H4. Any alternative clear mechanistic demonstration thought of by the authors would greatly improve the study.

Minor points:

1. "However, following initiation, transcription was rapidly attenuated downstream of TSSs in a region coincident with the CpG-rich region of the CGI where SET1 complexes bind (Fig.4g-h)."

The word "rapidly" implies a temporal property and should be reworded.

2. Figure 1m – label more clearly on the images that the depletion is combined SET1A/B. In addition, the green FISH dots are difficult to see on merged images, even on a computer screen. Please display individual channels in grayscale.

3. Figure 3 – The authors nicely show that a motif in SET1A/B is required for interaction with WDR82 and for supports gene expression in their reporter system. To solidify the authors conclusion that the observed lack of reporter expression is linked to the loss of interaction and not some other deficiency, the reciprocal experiment with WDR82 would be beneficial. I.e. perform this assay in a background of WDR82 with mutations that abrogate its interaction with SET1. We are aware that identifying the relevant residues in WDR82 might go beyond the time-frame of a revision, but if residues are already known such

experiment should be done. It will nail this important mechanistic aspect!

4. "Secondly, at SET1-independent genes, ZC3H4 283 enrichment is slightly biased upstream of gene promoters, coincident with the location of antisense 284 transcripts and consistent with a reported role for ZC3H4 in terminating upstream antisense 285 extragenic transcription (Fig.5c-d)."

Figure 5a-d – The ZC3H4 representative trace and the metaplot in Fig5a and Fig5d don't match. In the metaplot the signal before the TSS is only a tiny bit smaller than after, whereas there is quite a reduction before the TSS in the representative trace. This is important, because the metaplot could reflect two mutually exclusive scenarios – i.e. for each gene, ZC3H4 is either enriched upstream or downstream, but not both. Otherwise, there should be good individual examples of where the signal is found both up- and downstream. In either case, the authors should look at this and comment in the text. For Fig5d the authors highlight in the text that there is a binding bias upstream of gene promoters for SET1-independent genes, even though the metaplot curves in Fig5b and Fig5d (SET-independent) look barely distinguishable. Furthermore, both this upstream effect and the downstream binding bias for SET-dependent genes are characterized as "slight", even though the increased binding downstream of the TSS is more pronounced. While this doesn't change the overall conclusions of the study, statements relating to these data should be amended to describe the observations more accurately.

5. Line 179 – "bulk western" should read "bulk western blot".

6. Line 632 – Proteinase K digestion was performed at 45C. Most commonly, this is done at 55C, especially for short incubations such as 1h. Is this a typo, or is 45C correct?

Reviewer #3 (Remarks to the Author):

In this study, Hughes and colleagues report a catalytic activity-independent role of the SET1 complex in preventing early termination by WDR82-ZC3H4 at lowly expressed CpG island containing genes. The authors found that loss of SET1A and even more so the combined loss of SET1A/B reduced expression of low-to-moderately transcribed genes in a manner that did not correlate with loss of H3K4me3. Consistent with this observation, they found that artificial tethering of a catalytically inactive SET1 sufficed to increase transcription of a reporter gene and that this activity was dependent on the interaction with WDR82 which was mediated by a short linear motif. Reduced expression of SET1-dependent genes was associated with reduced Pol II occupancy inside genes with unmodified levels at their 5' ends, a finding consistent with increased premature intragenic termination. Critically, these effects were completely counteracted by the depletion of ZC3H4. These data led the authors to propose that while the ZC3H4-WDR82 complex acts unopposed to terminate transcription at enhancers and at promoter-divergent transcription units, it is efficiently neutralized by the SET1 complex at low-to-moderately expressed CpGi-containing genes, with highly expressed genes being instead constitutively resistant to termination.

Overall, the study provides a conceptually solid model, strongly supported by experimental data, that clarifies the interplay between two central machineries regulating transcription.

There are a few issues, mainly of minor relevance, that the authors may wish to consider to improve their study.

1. The data on SET1 tethering to the reporter gene in Fig. 2 and 3 clearly support a catalytic

activity-independent effect of SET1 on transcription and they show that, based on the effect of the DPR motif mutant, this effect is WDR82-dependent. However, they do not prove that this effect has anything to do with the prevention of termination, and thus with ZC3H4. It is possible that the effects observed here may merely reflect WDR82-mediated interactions with Pol II rather than the prevention of termination.

2. The interpretation of the ZC3H4 ChIP-seq data probably exceeds what can be rigorously extracted from the data. As ZC3H4 peaks appear to be very broad, it would be opportune to interpret more cautiously their relationship to the much narrower Pol II peaks and in particular the slight differences observed in the metaplots between SET1-dependent and independent genes (lines 281-290, figure 4A-D).

3. The effects of ZC3H4 depletion on protein coding genes were previously reported (ref. 78), with the CpGi-containing ZC3H4 gene being one example of such regulation. The current study, however, has the merit to provide a conceptual framework for such effects. What remains unclear is whether the magnitude of the effects of ZC3H4 depletion observed at genes is comparable to, lower or higher than that observed at extragenic regions.

4. While I agree with the main points raised in the discussion, in particular those relative to the unclear regulatory logic of the interplay between SET1 and ZC3H4, I do not agree with the message provided by the scheme in Figure 6h and in particular with the representation on the genic (right) side. This scheme hints at the idea of a free, non-WDR82-bound ZC3H4 which is somehow prevented from entering in contact with Pol II inside genes, which is not what ChIP-seq data show. One more reasonable model is that the many CTD repeats may accommodate multiple complexes via WDR82-mediated interactions with Ser5P-CTD and that the integration of different signals eventually determine the output.

5. The observation that ZC3H4/WDR82 effects are selectively antagonized by SET1 at genes while they are unopposed at extragenic regions is not reported in the abstract. As this represents a major conceptual and mechanistic aspect of this study, it should be properly highlighted.

We thank the reviewers for their very supportive comments and suggestions to improve our manuscript. We have now carried out a series of new experiments and analyses to address these constructive comments and have updated the main text and figures accordingly (altered text is highlighted red in the revised manuscript). Below we have provided a point-by-point response (blue text) to the reviewer's comments (black text). We believe these new experiments, analysis, and revisions have substantially improved the manuscript, and therefore we thank the reviewers for their time and extremely helpful input.

Reviewer #1 (Remarks to the Author):

In recent years, premature termination of transcription (PTT) has emerged as a prevalent process in human cells. In this paper, Hughes et al show that SET1 protects thousands of genes from PTT via the recently discovered termination factor, ZC3H4. Using the dTAG degron, they start by showing that 2000-odd genes show reduced mRNA levels after the rapid loss of SET1A. Fewer are affected by SET1B loss whereas co-depletion of SET1A and B affects the most genes. Basically, it looks like SET1A is more important than SET1B for this effect. The transcripts affected have CGI promoters and tend to be lowly expressed. Despite SET1A being widely studied as a histone methyltransferase, H3K4 methylation was essentially unaffected by its elimination. Furthermore, its promotion of gene transcription, in a tethered function assay, is unimpacted by mutation of its methyltransferase domain. Instead, it is proposed that SET1A promotes transcription by interaction with WDR82. This prevents the termination factor, ZC3H4, in complex with WDR82, from causing PTT at these genes. Consistently, transcriptional downregulation of SET1A-affected genes is reversed by co-depletion of ZC3H4. Overall, I found this to be an interesting and well-written paper - suitable for publication should the authors address the concerns/comments below.

We thank the reviewer for their supportive remarks and for their comments aimed at improving the manuscript.

1. Fig 1 shows that 2000-odd transcripts show reduced levels after rapid (2hr) SET depletion. This experiment is described as total RNA seq. If so, doesn't this mean that the SET1-affected transcripts must have very short half lives (for a steady state reduction to be seen after a relatively short period of time)? This seems like an interesting observation worthy of discussion unless I misunderstand it.

We thank the reviewer for raising this important question. Our cRNA-seq analysis was carried out on total RNA that was depleted of rRNA. These assays measure total transcript levels, comprising a mixture of unspliced and spliced transcripts from both the nucleus and the cytoplasm. The effects we can capture in cRNA-seq after depletion of SET1 complexes will therefore be influenced by both the production of new transcripts and the turnover of existing transcripts. As such, changes observed in cRNA-seq analysis at 2 hrs after SET1 complex depletion could be biased towards genes with less stable transcripts as suggested by the reviewer. To interrogate this possibility, we have examined the half-lives of genes (from ¹) with significantly reduced expression, and those that are unchanged, as measured by cRNA-seq analysis. As is evident in Supplementary Fig. 4e, and also shown below for the benefit of the reviewer (Reviewer Figure 1), we find that genes with reduced expression in our cRNA-seq analysis tend to have shorter half-lives than those that are not significantly changed. However, importantly, when we carry out the same half-life analysis on genes that are significantly reduced in transcription, as measured by cTT-seq, which captures ongoing transcription, we see very little difference in the mean half-lives of the genes with reduced transcription and those genes that are unchanged. In the revised manuscript we now discuss the limitation of transcript half-life on cRNA-seq analysis when we introduce the cTT-seq analysis on lines 238-246:

'However, in comparison to cRNA-seq, cTT-seq identified more genes that had reduced transcription and these reductions were larger in magnitude (Supplementary Fig.4d). This difference likely arises from the fact that cRNA-seq interrogates total RNA levels which are influenced by both the rate of transcript production and degradation. Indeed, genes that were reduced in expression after SET1 complex depletion, as measured by cRNA-seq, tended to have shorter transcript half-lives than unchanged genes (Supplementary Fig.4e). In contrast, genes with reduced transcription, as measured by cTT-seq, were not subject to this bias. Therefore, we conclude that cTT-seq captures the primary influences of SET1 complexes on gene transcription.'

Reviewer Figure 1

Box plots showing the transcript half-lives of genes that have either reduced or unchanged expression following SET1 protein depletion, as measured by cRNA-seq (left panel) or cTT-seq (right panel) experiments.

2. The lack of effect of SET1 loss on H3K4 is very interesting given the long-held view that it is a methyltransferase. The TetO tethering experiment is also a very useful way of demonstrating that this activity is unlikely to underpin the effect on transcription. This lack of effect on methylation (and of this on transcription) seems important and goes against the grain. As such, I think an additional experiment is warranted whereby mutated SET1 is used to rescue the effects of SET1A depletion on transcription. This need not be a full sequencing experiment since the authors have already identified thousands of SET1 transcription targets, a sample of which could be used for this experiment.

We thank the reviewer for suggesting this interesting experiment. To address this point, we have used genome engineering to develop a tamoxifen-inducible ESC system where we can convert the endogenous SET1A locus from expressing wild type SET1A into a catalytically inactive form of the protein (see Reviewer Figure 2a and ²). We have previously found that this endogenous conversion approach is essential for overcoming the numerous complications (e.g. dominant negative effects and variable transgene expression levels) that are associated with attempts to rescue protein depletion via exogenous transgene expression. Importantly, using this approach we could efficiently induce expression of the catalytically inactive SET1A transcript (Reviewer Figure 2b). However, despite the catalytically inactive transcript being produced at a similar level to the wild type transcript, cells induced to express the catalytically inactive transcript had dramatically reduced levels of SET1A protein (Reviewer Figure 2c-d). We presume this is due to the mutant protein being less stable, which would be consistent with findings that the SET1A-related protein MLL4 also displays reduced protein

levels when catalytically inactive³. It is currently unclear why disrupting catalysis affects SET1A protein levels. However, given that the catalytically inactive form of SET1A displays dramatically reduced protein levels, this technical limitation means we are unable to definitively test the contribution of SET1A catalysis to transcription and gene expression. In future work, we will need to understand why SET1A levels are reduced after mutating its catalytic domain, and to identify strategies to retain wild type levels of the mutant protein in order to definitely test the contribution of catalysis to transcription and gene expression. To ensure that it is clear to the reader that we have limited our interpretation of the catalytic mutant experiments to the reporter assays, we have now clearly stated on lines 199-201 in the revised manuscript that:

'alterations in gene expression observed after SET1 protein depletion do not primarily manifest from a loss of H3K4me3 and that SET1A can support reporter gene expression independently of its methyltransferase activity.'

Reviewer Figure 2

(a) The endogenous *Set1a* alleles were homozygously engineered to contain a conditional point mutant (CPM) cassette flanked by doubly inverted LoxP sites, as illustrated (also see²). Upon tamoxifen (OHT) treatment, ERT2-CRE (expressed from the *Rosa26* locus) flips the orientation of the CPM cassette, causing the *Set1a* gene to utilize exons encoding the catalytic mutations in the SET domain (N1655S and Y1693F).

(b) Quantitative RT-PCR analysis of the *Set1a* conditional point mutant cell line using primer sets specific for either the wild type catalytic exons of *Set1a* (red, WT) or the catalytic mutant exons of *Set1a* (purple, CAT MUT). Following tamoxifen (OHT)-induced flipping of the CPM cassette, the *Set1a* transcript incorporates the catalytic mutant exons.

(c) Quantitative RT-PCR analysis of the *Set1a* conditional point mutant cell line using a primer that detects all *Set1a* transcript (WT and CAT MUT). This shows that *Set1a* transcripts incorporating the WT or catalytic mutant exons are expressed at similar levels.

(d) Western blot of the SET1A CPM cell line. Following tamoxifen (OHT) treatment, SET1A protein levels are significantly reduced.

3. Related to the above (minor point), could other H3K4 transferase activities substitute the loss of SET1A/B even after rapid loss?

We believe this is a plausible explanation for the very modest decrease in H3K4me3 observed after SET1A/B depletion. This would be consistent with previous findings in longer term perturbation

experiments, where the additional disruption of MLL2 caused more pronounced effects on H3K4me3 than the deletion of the SET1A SET domain alone ⁴. We envisage that SET1A/B do contribute to H3K4me3, but that MLL1/2 or other H3K4 methyltransferases compensate for their depletion. In future experiments, combinatorial depletions with additional known H3K4 methyltransferases will be needed to address this important point. However, the very modest effect on H3K4me3 caused by SET1A/B depletion fortuitously allowed us to characterise the profound and seemingly H3K4me3-independent role that SET1A/B play in counteracting premature transcription termination, which is the focus of this manuscript.

4. Figure 3 clearly shows that just the WDR82 binding part (NTD) of SET1A is sufficient for its transcriptional effects (to drive luciferase expression). Presumably, this means the effect in cells could be due to some other (WDR82-interacting?) factor rather than SET1A that is pulled in by WDR82-SET1A. I say this because the DPR fragment that cannot bind WDR82 is unable to drive luciferase expression. Can this be discussed/speculated on or do the authors have an idea of what it might be?

We thank the reviewer for bringing up this important point. Based on the DPR mutant experiments we believe that the effects we observe on expression depend on the capacity of the SET1A N-terminal domain to interact with WDR82. We agree with the reviewer that we cannot exclude the possibility that that another unknown factor that interacts with WDR82 also contributes to the effects we observe on expression. However, our biochemical purifications of SET1A complexes from ESCs (unpublished) have not revealed any additional stoichiometric complex components beyond those already known and reported to interact with other regions of SET1A. Therefore, we believe it is likely that the effect we observe could be mediated directly via the influence of WDR82 and possibly through its characterised interactions with RNA Pol II. Nevertheless, we have now updated the text in the revised manuscript to highlight the possibility other unknown WDR82-dependent protein interactions could be involved in enabling gene expression on lines 223-226 as follows:

‘Therefore, a highly conserved DPR motif in the N-terminal domains of SET1 proteins is required for their interaction with WDR82 and this interaction can support gene expression either directly or through additional protein interactions.’

5. ZC3H4 depletion also causes upregulation of many sense transcripts. What is the overlap between this set and those affected by SET1A loss? Would it be predicted that they would not overlap given the proposed role of SET1A in suppressing ZC3H4 effects? FigS6 has some of this information but additional information for the SET1-independent genes would be more complete and may add to the data shown for ZC3H4.

To address this point, we have compared the levels of transcription of sense transcripts (genes) that are significantly reduced when SET1 complexes are depleted with those that are increased when ZC3H4 is depleted. As envisaged by the reviewer, the majority of these affected genes do not overlap. However, importantly, the genes that are unique to SET1 complex depletion are more highly transcribed than the ZC3H4-dependent genes. We envisage that this is because at the SET1-dependent gene set, the SET1 complex predominates in limiting ZC3H4 activity. Conversely the genes that are uniquely affected by ZC3H4 complex depletion tend to be very lowly transcribed, while the genes that overlap between the two depletions appear to have an intermediate expression between the SET1- and ZC3H4-specific subsets. These observations are consistent with the idea that the ZC3H4 complex may function pervasively throughout the genome to terminate transcription at more lowly transcribed regions, and that SET1 complexes are enriched at actively transcribed genes, where they protect low to moderately transcribed genes from the effects of ZC3H4. To make this point we have now added this comparison as a Venn diagram in Supplementary Fig.S6b of the revised manuscript (for

convenience, please also see also Reviewer Figure 3) and described these results on lines 319-330 of the revised manuscript as follows:

'A shared feature amongst genes regulated by ZC3H4 and SET1 proteins is that they tend to be low to moderately transcribed (Fig.4 and Fig.5). Therefore, we were curious whether ZC3H4 regulates the transcription of SET1-dependent genes. Consistent with this possibility, after depletion of ZC3H4 we observed a small increase in transcription at SET1-dependent genes (Fig.6a-b, Supplementary Fig.6a and 6e). Interestingly, SET1-dependent genes with significantly increased transcription after ZC3H4 depletion were more lowly transcribed than other SET1-dependent genes, whereas the genes that were only affected after ZC3H4 depletion had even lower levels of transcription (Supplementary Fig.6b). These observations are consistent with the idea that ZC3H4 functions pervasively throughout the genome to drive PTT at lowly transcribed regions and that this activity can influence SET1-dependent genes. Furthermore, if SET1 proteins oppose ZC3H4-dependent PTT as we hypothesise, this would imply that SET1 activity must not be sufficient to completely counteract the influence of ZC3H4, particularly at the most lowly transcribed genes.'

Reviewer Figure 3

A Venn diagram illustrating the overlap between genes that have significantly reduced transcription following SET1A/B depletion (dTAG-SET1A/B: DOWN) and genes that have significantly increased transcription following ZC3H4 depletion (ZC3H4-dTAG: UP) (bottom panel). The Box plot illustrates the level of transcription in untreated dTAG-SET1A/B cells for genes uniquely affected by SET1A/B depletion, overlapping genes, and genes uniquely affected by ZC3H4 depletion (top panel).

6. Related to 5, are those genes affected by ZC3H4 less likely to have strong CGIs (i.e. the opposite relationship to the SET1 affected)?

As suggested by the reviewer we have looked at the genes with increased transcription after ZC3H4 depletion and we find that they tend to have a lower CpG density and SET1A occupancy compared to unchanged genes. We believe these features, coupled with transcription level, could contribute to ZC3H4 sensitivity and also explain why SET1 complexes do not fully counteract ZC3H4 activity at these genes. To highlight this important point, we have included this analysis in Figure 5j (see Reviewer Figure

4 below for convenience) and drawn attention to it in the revised manuscript on lines 309-310 as follows:

‘Interestingly, genes that were most susceptible to ZC3H4 loss were typically low to moderately transcribed, and had low CpG density and SET1A occupancy (Fig.5i-j).’

Reviewer Figure 4

Metaplots comparing CpG density and SET1A levels at the TSSs of CGI-associated genes that are increased in transcription after 2 hours of ZC3H4 depletion (n=1653) and those that are unchanged (n=12667).

7. ChIP-seq shows ZC3H4 presence at promoters of SET1-dependent and independent genes whereas the model in Fig 6 depicts that SET1 opposes the recruitment of ZC3H4 to the promoters of SET1-dependent genes. In figure 5C there is more ZC3H4 recruited to the coding direction of the SET1 dependent genes vs the independent set. Isn't this inconsistent with the model as drawn? This suggests that ZC3H4 is recruited to promoters regardless of whether they are susceptible to SET1 effects, which isn't what the model implies.

We thank the reviewer for bringing up this important point. The reviewer has interpreted our observations in the way we had intended and, in retrospect, we appreciate that these important findings may not have been clearly enough described in the initial submission and in the associated model figure. What we observe in our genomic experiments is that ZC3H4 engages with most regions of the genome where transcription is initiating (transcribed promoters, transcribed upstream antisense regions, and transcribed enhancers) in agreement with previous findings^{5,6}. There is a slight bias in the directionality of ZC3H4 binding at SET1-dependent genes, but we should note that this is modest and binding of ZC3H4 is evident at both the sense and antisense initiation events. We predominantly find ZC3H4 suppresses transcription at lowly transcribed regions including lowly transcribed genes, enhancers, and upstream antisense transcripts. We speculate that lowly transcribed regions are more sensitive to termination by ZC3H4 due to a paucity of transcription activation signals and/or transcription-associated processivity factors that might counteract ZC3H4 activity at more highly transcribed genes. As such, we envisage a model whereby ZC3H4 pervasively engages with regions of the genome where transcription is initiating and attempts to terminate transcription. This primarily influences productive elongation from lowly transcribed regions, especially upstream antisense transcripts and enhancer RNAs that are not protein-coding and, in many cases, unlikely to be functional. However, this termination activity could also impinge on lowly transcribed protein-coding genes. We envisage that SET1 complexes engage on the protein-coding, CpG-dense side of the promoter to distinguish genic from non-genic transcription and counteract

ZC3H4-dependent transcription termination. In doing so, SET1 complexes appear to ensure that lowly transcribed protein-coding genes are protected from premature termination by ZC3H4.

A similar point was also raised by reviewer three with respect to our model figure. Therefore, to ensure this model is clear to readers, we have clarified the description of these findings in the revised figure legend text, ensured that the very modest differences in binding of ZC3H4 at SET1-dependent and SET1-independent genes is appropriately qualified (lines 294-296), and have updated the model figure (Figure 6h, and Reviewer Figure 5) to make it clear that SET1 complexes appear to counteract ZC3H4 activity on the protein-coding side of CpG island-associated gene promoters.

Reviewer Figure 5

A cartoon illustrating a model whereby WDR82-containing SET1 complexes bind to CpG-dense regions in CGIs downstream of TSSs to enable genic transcription by counteracting premature transcription termination by WDR82-containing ZC3H4 complexes. The defined mechanism through which SET1 complexes counteract the function of ZC3H4 complexes remains to be determined, but this likely involves both SET1 and ZC3H4 complexes interacting with the CTD of RNA Pol II and the integration of their distinct activities determining the effect on transcription. In contrast, extragenic transcription that emanates from regions lacking SET1 complex occupancy is subject to termination by WDR82-containing ZC3H4 complexes. In this model, CGIs and SET1 complex occupancy would distinguish genic from extragenic transcription and protect genic transcription from premature transcription termination to enable gene expression.

8. In yeast, WDR82 links SET1 to Pol II. Could this be important in the present paper as an additional/alternative mode of recruitment vs CGIs?

We have previously shown that multivalent interactions with non-methylated CpG DNA and H3K4me3 via the CFP1 component of the SET1 complex are primarily responsible for specifying SET1A occupancy at actively transcribed CGI-associated gene promoters in ESCs⁷. The capacity of the SET1 complex to interact with RNA Pol II via WDR82 appears to be conserved in vertebrates^{8,9}. While we envisage that it is the SET1/WDR82 complex's binding to CGI chromatin and its interaction with RNA Pol II that counteracts the termination activity of ZC3H4/WDR82 complexes (Figure 6h), we agree with the reviewer that the interaction between SET1/WDR82 could also play an important role in initially

guiding the complex to actively transcribed CGI-associated gene promoters. We have now highlighted this possibility in the discussion of the revised manuscript on lines 403-405 as follows:

'SET1 complexes generically associate with CpG-rich DNA just downstream of TSSs on the genic side of transcribed CGI-associated genes through multivalent chromatin binding mechanisms and may also initially sense transcription at these sites via WDR82 binding to Ser5P on the RNA Pol II CTD^{12,24-30}.'

Reviewer #2 (Remarks to the Author):

The authors present a manuscript that attempts to delineate the mode of action of SET1 complexes in transcription regulation. Throughout the study, rapid depletions are used to focus on direct targets and limit pleiotropic effects. This allows the authors to separate the effects of H3K4me3 loss and loss of SET1 proteins itself. Surprisingly, this reveals an important, potentially methyltransferase-independent role of SET1 in regulation of lowly-expressed genes. The redundancy of methyltransferase activity for this regulation is elegantly confirmed by a series of in vivo reporter assays, but not at the genome-wide level as the rest of the study. The authors show data supporting the view that SET1 antagonizes ZC3H4-dependent premature transcription termination. This is supported by examining the effects of ZC3H4 depletion, and simultaneous depletion of ZC3H4 and SET1.

The work is original, overall well executed, clearly presented, and would be of interest to several fields of research in molecular biology and biomedicine. Nevertheless, the manuscript would greatly benefit from some additional experiments, analyses, clarifications, and additional mechanistic examination. Please see the point-by-point comments for details.

We thank the reviewer for their careful consideration of our manuscript and for making a series of important suggestions which have clarified several of our conclusions and generally improved the manuscript. We have detailed these in the following point-by-point response to the reviewer comments.

Major points:

1. "The addition of the dTAG did not affect SET1A protein levels (Fig.1a), SET1A complex formation (Supplementary Fig.1a), or its preferential localisation to CpG-rich regions downstream of TSSs at expressed CGI-associated gene promoters"

To properly support the claim that tagging SET1A with FKBP12 doesn't affect preferential localisation, it would be necessary to show a side-by-side ChIP-seq analysis of tagged and untagged line.

We thank the reviewer for highlighting this oversight and we have now carried out new analysis to address this point. As illustrated in Reviewer Figure 6 below, we have now compared cChIP-seq signal for T7-dTAG-SET1A with ChIP-seq signal for endogenous SET1A¹⁰. As is evident from the genomic snapshots, heatmaps, and scatter plots, the T7-dTAG-SET1A cChIP-seq signal is highly correlated with the endogenous SET1A ChIP-seq signal. Due to space constraints in the supplementary figure we have not been able to include this entire analysis in the revised manuscript. However, to support our initial claim we have included the scatter plot of T7-dTAG-SET1A and endogenous SET1A ChIP-seq signal and the correlation values corresponding to this data in Supplementary Fig.1c of the revised manuscript.

Reviewer Figure 6

(a) A genomic snapshot showing ChIP-seq signal for T7-dTAG-SET1A (in untreated cells and cells treated with dTAG13 for 2 hours) and endogenous SET1A¹⁰.

(b) Heatmaps illustrating ChIP-seq signal for T7-dTAG-SET1A and SET1A at CGIs that are within 1.5 kb of a TSS (n=15259).

(c) A scatter plot illustrating T7-dTAG-SET1A and SET1A ChIP-seq signal at CGIs that are within 1.5 kb of a TSS (n=15259). The spearman correlation and R^2 value are indicated.

The authors claim that SET1A complex formation is not affected, but IP efficiencies in FigS1A appear slightly reduced, including the bait protein. Could the authors provide a more even blot?

In our experience, the total amount of material immunoprecipitated between extract samples can vary slightly between experiments, as is evident in Supplementary Fig.1a. However, the central point of this experiment was to illustrate that the addition of the dTAG to SET1A does not disrupt SET1A complex formation, which is evident from the fact that dTAG-SET1A immunoprecipitates SET1A complex components. To clarify this point, we have edited this statement in the revised manuscript on lines 111-114 as follows:

'The addition of the dTAG did not affect SET1A protein levels (Fig.1a), disrupt SET1A complex formation (Supplementary Fig.1a), or its preferential localisation to CpG-rich regions downstream of TSSs at expressed CGI-associated gene promoters (Fig.1c-d and Supplementary Fig.c)'

2. "Interestingly, changes in gene expression were less pronounced at later time points after SET1A depletion, suggesting that additional mechanisms may compensate for its depletion over time (Supplementary Fig.1e)."

For clarity, authors should specify what "additional mechanisms" refers to. For instance, do they suggest the effect is seen due to technical reasons, such as selection for cells with poorer depletion in cell culture, or some other cell-intrinsic biological phenomenon, or both.

We thank the reviewer for raising this point. In our original submission we simply referred to the possibility that 'additional mechanisms' may compensate for the depletion of SET1 proteins over time as we do not have direct experimental evidence supporting what those mechanisms might be. However, we have not observed any apparent waning of SET1 protein depletion over time (Reviewer Figure 7), so we believe these 'additional mechanisms' may be some alternative secondary adaptation of the cells that leads to the observed effects on gene expression. A key reason for showing the data

from later time points was to highlight the importance of analysing early time points to capture the primary influences of SET1 proteins on gene expression and transcription. Given our lack of understanding about the mechanism(s) leading to this apparent compensation at later time points, we would prefer to avoid speculating any further. However, we have now slightly reworded this sentence in the revised manuscript on lines 125-128 to highlight the reviewers point as follows:

'Interestingly, changes in gene expression were less pronounced at later time points after SET1A depletion, suggesting that additional mechanisms or cellular adaptations may compensate for its depletion over time (Supplementary Fig.1f).'

Reviewer Figure 7

Western blot analysis of nuclear extract from the dTAG-SET1A/B cell line after 0, 2, 4 and 24 hours of dTAG13 treatment. The levels of SET1B and SET1A protein are shown. An extract from wildtype (WT) ESCs is shown for comparison, and western blot with a SUZ12-specific antibody serves as a loading control.

3. Figure 1k and Figure S1h+j. It is clear that SET1A/B depletion predominantly results in gene downregulation, but the authors should include the upregulated gene category in these analyses, particularly because this category is included in Figure 2c.

As suggested by the reviewer, we have now added the genes that have increased expression to the analysis in Figures 1k and Supplementary Fig.1h and 1j (now Supplementary Fig.1i and 1k) of the revised manuscript (see Reviewer Figure 8 below for convenience).

Reviewer Figure 8

(a) A box plot showing the expression level in untreated cells (UNT RPKM) for expressed genes with reduced expression (Reduced, n=2544) following 2 hours of

SET1A/B depletion, and those that are increased in expression (Increased, n=495) or unchanged (Unchanged, n=9989)

(b) Metaplots comparing CpG density, SET1A levels, and H3K4me3 levels at the TSSs of CGI-associated genes that are reduced in expression after 2 hours of SET1A/B depletion (n=2571) and those that are increased in expression (n=609) or unchanged (n=11258).

(c) A bar plot showing the percentage of genes associated with CGIs comparing all genes (n=20633) to those that are reduced in expression after 2 hours of SET1A/B depletion (n=2928), genes that are increased in expression (n=745) and those that are unchanged (n=16960).

4. Figure S3A – The NTD of SET1B appears to have an effect several fold higher than the NTD of SET1A, at similar expression levels. The authors make no comment on this, and currently the phrasing in the text implies the effects to be equivalent. One potential explanation for this is that SET1B-NTD (and therefore probably the protein itself) is a stronger WDR82 interactor. The manuscript would benefit from at least some test whether this is the case.

Based on our preliminary unpublished *in vitro* biochemical analysis, we have not observed any differences in how SET1A and SET1B interact with WDR82. We believe a careful, quantitative examination of these biochemical interactions will be important and we aim to do this in the context of future detailed *in vitro* biochemical and structural work which goes beyond the scope of our current study. The reason we included the SET1B NTD in the reporter gene analysis in Supplementary Fig.3 was simply to test whether this region of both SET1A and SET1B could support gene expression. However, to make it clear to the reader that there were differences in the magnitude of the effects on expression when the SET1A and SET1B NTDs were examined, we have edited the text in the revised manuscript on lines 214-216 to read as follows:

‘Furthermore, the equivalent NTD of SET1B was also sufficient to support gene expression, and it did so more efficiently than the NTD of SET1A, indicating that this activity is conserved amongst SET1 paralogues (Supplementary Fig.3a-b)’

5. A major conclusion of this study is that SET1 functions independently of its methyltransferase activity, which is supported by the SET1A tethering reporter assay. To indeed generalise such conclusion, it is imperative to show this genome-wide. Therefore, the authors must demonstrate that the SET1A-depletion phenotype can be rescued by ectopic expression of the inactive form of the protein.

As described in detail in the response to Reviewer 1 point 2, and illustrated in Reviewer Figure 2, we have now developed a system to induce the conversion of endogenous SET1A into a catalytically inactive form. Despite this producing similar levels of mRNA to the wild type allele, the loss of catalysis causes a major reduction in SET1A protein levels. We presume this is due to the mutant protein being less stable, which would be consistent with findings that the SET1A-related protein MLL4 also displays reduced protein levels when catalytically inactive³. Given that the catalytically inactive form of SET1A displays dramatically reduced protein levels in cells, this technical limitation means we are unable to definitively test the contribution of catalysis by SET1A on transcription and gene expression genome-wide. To address this question in future work will require understanding why SET1A levels are affected after mutating the catalytic domain and to identify strategies to retain wild type levels of the mutant protein. To ensure that it is clear to the reader that we have limited our interpretation of the catalytic

mutant experiments to the reporter assays, we have now clearly stated on lines 198-201 in the revised manuscript that:

‘Together, our histone modification analysis and tethering experiments suggest that alterations in gene expression observed after SET1 protein depletion do not primarily manifest from a loss of H3K4me3 and that SET1A can support reporter gene expression independently of its methyltransferase activity.’

6. Same critique as raised in point 5 applies to Figure 3. Here it must be demonstrated genome-wide that ectopic expression of the DPR/AAA mutant does not rescue SET1A depletion.

This is an excellent suggestion by the reviewer. Before initiating more complicated cell engineering experiments aimed at rescuing SET1A function with the WDR82-binding mutant (see point 5 above for description of the CPM system), we wanted to examine how the absence of an interaction with WDR82 might influence the function of SET1A. To this end, we created a new cell line where we homozygously engineered a dTAG into the endogenous *Wdr82* gene. Treatment of the WDR82-dTAG cell line with dTAG13 resulted in a complete depletion of WDR82 within 2 hours. When we examined SET1A by western blot after WDR82 depletion, we observed a major reduction in SET1A protein levels (Reviewer Figure 9). Unfortunately, we believe the requirement for WDR82 in maintaining SET1A protein levels precludes us from addressing at the genome-scale whether the effects of SET1A depletion can be rescued by a mutant version of SET1A that cannot bind to WDR82. To address this question in future work it will be important to understand why the levels of SET1A are reduced in the absence of WDR82.

Reviewer Figure 9

A WDR82-dTAG ESC line was generated and nuclear extract prepared from both untreated cells and cells treated with dTAG13 for 2 hours. Western blot analysis confirmed depletion of WDR82 following dTAG treatment. Depletion of WDR82 caused reduced levels of SET1A protein.

7. The findings described in Figure 6 and associated text suggest that loss of ZC3H4 is dominant in relation to the loss of SET1 when it comes to transcriptional output of SET1-dependent genes. The authors conclude that therefore the primary function of SET1 in this context is to antagonize ZC3H4. However, degradation of ZC3H4 on its own results in a minor increase in transcription. A possible explanation for that is that WDR82 is destabilised upon ZC3H4 depletion. In this case, potential direct activity of SET1 promoting transcription elongation would be lost, possibly partially compensating for the loss of ZC3H4. The authors should address this possibility by including a western blot for WDR82 under the same conditions as in FigS6B.

To test whether depletion of ZC3H4 causes destabilisation of WDR82 as suggested by the reviewer, we have now analysed the levels of WDR82 by western blot after a 2 hour depletion of ZC3H4. As illustrated in Supplementary Fig.5d of the revised manuscript and Reviewer Figure 10 below, we

observed no effects on WDR82 levels after ZC3H4 depletion. We have now highlighted this important point to the reader in the context of the revised manuscript on lines 298-299 as follows:

'dTAG13 treatment resulted in a near-complete depletion of ZC3H4 within 2 hours, with no effect on the levels of its interaction partner WDR82 (Fig.5d and Supplementary Fig.5d)'

Reviewer Figure 10

Western blot analysis of ZC3H4-dTAG cells showing that two hours treatment with dTAG13 causes complete depletion of ZC3H4, while levels of WDR82 are unchanged.

The authors should also show whether the chromatin binding of ZC3H4 is dependent on SET1 and vice versa.

Our previous work has demonstrated that SET1A binding to chromatin relies primarily on the CFP1 protein and its capacity to utilise multivalent interactions to engage with CpG island chromatin ⁷. However, we agree with the reviewer that it would be interesting to know whether the chromatin binding of ZC3H4 is influenced by SET1 complexes. To address this point, we have now depleted SET1 complexes in the dTAG-SET1A/B line and carried out ChIP-seq for ZC3H4. This demonstrated that ZC3H4 binding was modestly reduced in the absence of SET1A/B. This observation is in line with modest reductions in promoter-associated RNA Pol II (Figure 4d) that are also observed when SET1A/B are depleted. This leads us to speculate that SET1 complexes likely influence ZC3H4 activity at target sites, as opposed to regulating its binding. We have included this new ChIP-seq experiment and analysis in Supplementary Fig.6i (and Reviewer Figure 11 for convenience) and described this important finding in the context of the revised manuscript on lines 344-349.

'Importantly, this antagonism does not appear to manifest from direct physical competition for binding to RNA Pol II or WDR82 as depletion of SET1 complexes did not lead to increased binding of ZC3H4 at TSSs and WDR82 is in excess to SET1 and ZC3H4- complexes in ESCs (Supplementary Fig.6h and 6i). As such, we envisage that both SET1 and ZC3H4 complexes interact with the CTD of RNA Pol II and the integration of their distinct activities determines the effect on transcription (Fig.6h)'

Reviewer Figure 11

A metaplot illustrating ZC3H4 ChIP-seq signal at the TSSs of all genes (n=20633) in the dTAG-SET1A/B cell line that is either untreated (UNT) or treated with dTAG13 for 2 hours.

8. In the discussion, the authors propose that SET1-WDR82 can antagonize the activity of ZC3H4-WDR82. While the functional antagonism is described, it is not made clear how it is achieved mechanistically. The manuscript would benefit greatly from further mechanistic investigation. The simplest mechanism, which is also hinted in the model in Fig6h would be that SET1 and ZC3H4 compete for WDR82 binding. One way this could be tested is by a series of co-IP experiments, wherein SET1 or ZC3H4 are depleted, and the other factor immunoprecipitated. If there is competition, the amount of WDR82 pulled down should increase in both conditions compared to non-depleted. Otherwise, overexpression of the NTD of SET1 might cause reduced pulldown efficiencies for both endogenous SET1 and ZC3H4. Any alternative clear mechanistic demonstration thought of by the authors would greatly improve the study.

One can envisage a number of potential mechanistic models to explain the antagonism between SET1/WDR82 and ZC3H4/WDR82 complexes. The model alluded to by the reviewer is plausible. However, in exploring this possibility in new biochemical experiments, we observe by size exclusion chromatography that there is actually a significant excess of uncomplexed WDR82 in ESCs. This suggests that WDR82 levels are not limiting for inclusion in either the SET1 or ZC3H4 complexes (or the PNUTS complex that forms a distinct termination complex that functions at the 3' end of genes) (now presented in Supplementary Fig.6h and Reviewer Figure 12 for convenience). Furthermore, in the context of the reviewer's helpful suggestion in point 7, we did not observe an increase in ZC3H4 binding when SET1 proteins were depleted. This finding is not consistent with a model that involves a direct physical competition for binding to WDR82. In fact, our depiction of ZC3H4 not being bound to WDR82 on the genic side of the gene in our model cartoon in Fig.6h of the original submission was an oversight. We had intended to illustrate ZC3H4 bound to WDR82, which we have now remedied in our revised model figure. In line with these findings, we have updated the text of the discussion to make it clear that we do not believe that competition for binding to WDR82 or RNA Pol II explains the antagonism between SET1/WDR82 and ZC3H4/WDR82 complexes as described above in response to reviewer point 7.

We are also very keen, like the reviewer, to understand the detailed mechanisms through which SET1/WDR82 complexes antagonise ZC3H4/WDR82 complexes. However, we believe the type of '*clear mechanistic demonstration*' sought by the reviewer will first require a number of poorly understood yet basic mechanistic questions about ZC3H4/WDR82 function to be answered. First and foremost, it will be necessary to define the mechanisms through which ZC3H4/WDR82 causes premature transcription termination. We envisage that this will require detailed biochemical and structural

interrogation using *in vitro* reconstituted systems where the influence of ZC3H4/WDR82 on transcription by RNA Pol II can be directly examined. Once the mechanisms that enable transcription termination by ZC3H4/WDR82 are determined, the addition of reconstituted SET1 complexes to these *in vitro* reactions could be used to define how the SET1 complex influences termination by ZC3H4. While we have initiated these lines of investigation, this remains a major undertaking that goes well beyond what is possible within the scope of our current manuscript. Nevertheless, we believe the discoveries described in our manuscript represent a major advance in our understanding of how CpG islands and SET1 complexes regulate transcription. Furthermore, we identify a new regulatory link between the SET1 complexes and control of premature transcription termination by ZC3H4 complexes. Together, we believe this represents a major conceptual advance in our understanding of the mechanisms that control gene transcription and expression.

Reviewer Figure 12

Nuclear extract from mouse ESCs was subjected to size exclusion chromatography and the resulting fractions probed for WDR82, SET1A, ZC3H4, and PNUTS by western blot. This revealed that a significant excess of WDR82 is present in ESCs that is not complexed with SET1A, PNUTS or ZC3H4 (e.g. see fractions 37-43).

Minor points:

1. "However, following initiation, transcription was rapidly attenuated downstream of TSSs in a region coincident with the CpG-rich region of the CGI where SET1 complexes bind (Fig.4g-h)."

The word "rapidly" implies a temporal property and should be reworded.

We agree with the reviewer and have now removed the word 'rapidly' in the revised manuscript.

2. Figure 1m – label more clearly on the images that the depletion is combined SET1A/B. In addition, the green FISH dots are difficult to see on merged images, even on a computer screen. Please display individual channels in grayscale.

We thank the reviewer for this suggestion and have edited the title of Fig.1m accordingly to make it clear that these FISH experiments were performed in the dTAG-SET1A/B line. To make the FISH dots easier to see in the merged images, we increased the brightness and contrast of the dots while also changing the DAPI pseudo-colouring to red (see Fig.1m and Reviewer Figure 13a). We also displayed the individual channels in grayscale (Reviewer Figure 13b), though our feeling was that this didn't make the FISH dots any easier to see, while also causing the figure panel to occupy more space in an already busy figure.

Reviewer Figure 13

(a) Example images of smRNA-FISH for the *Mcat* gene in the *dTAG-SET1A/B* line, showing an untreated (UNT) cell and a cell treated with *dTAG13* for 2 hours. White spots correspond to *Mcat* RNAs and red corresponds to DAPI staining of DNA. The white scale bars correspond to 10 μm .

(b) A Greyscale version of (a) for comparison.

3. Figure 3 – The authors nicely show that a motif in SET1A/B is required for interaction with WDR82 and for supports gene expression in their reporter system. To solidify the authors conclusion that the observed lack of reporter expression is linked to the loss of interaction and not some other deficiency, the reciprocal experiment with WDR82 would be beneficial. I.e. perform this assay in a background of WDR82 with mutations that abrogate its interaction with SET1. We are aware that identifying the relevant residues in WDR82 might go beyond the time-frame of a revision, but if residues are already known such experiment should be done. It will nail this important mechanistic aspect!

We thank the reviewer for this suggestion. We have attempted to generate mutations in WDR82 that specifically disrupt its interaction with SET1 proteins. However, the compact and structured nature of WDR repeat proteins makes it difficult to be confident that such mutation(s) do not simply disrupt protein structure as opposed to specifically disrupting an interaction interface for SET1 proteins. We have initiated crystallisation trials with the goal of obtaining atomic level information about WDR82 and its interaction with SET1 NTD fragments. Should we be successful, we hope to use this information in the context of future biochemical studies to rigorously address this question.

4. "Secondly, at SET1-independent genes, ZC3H4 enrichment is slightly biased upstream of gene promoters, coincident with the location of antisense transcripts and consistent with a reported role for ZC3H4 in terminating upstream antisense extragenic transcription (Fig.5c-d)."

Figure 5a-d – The ZC3H4 representative trace and the metaplot in Fig5a and Fig5d don't match. In the metaplot the signal before the TSS is only a tiny bit smaller than after, whereas there is quite a reduction before the TSS in the representative trace. This is important, because the metaplot could reflect two mutually exclusive scenarios – i.e. for each gene, ZC3H4 is either enriched upstream or downstream, but not both. Otherwise, there should be good individual examples of where the signal is found both up- and downstream. In either case, the authors should look at this and comment in the text.

For Fig5d the authors highlight in the text that there is a binding bias upstream of gene promoters for SET1-independent genes, even though the metaplot curves in Fig5b and Fig5d (SET-independent) look barely distinguishable. Furthermore, both this upstream effect and the downstream binding bias for SET-dependent genes are characterized as "slight", even though the increased binding downstream

of the TSS is more pronounced. While this doesn't change the overall conclusions of the study, statements relating to these data should be amended to describe the observations more accurately.

We agree with the reviewer that the difference in the occupancy of ZC3H4 at SET1-independent and SET1-dependent genes is very modest. Nevertheless, we thought it relevant to draw attention to this as a potential difference in how ZC3H4 behaves at genes that have different transcription levels and sensitivities to SET1 depletion. The genomic snapshots in Figure 5c are extreme examples and we agree in hindsight they do not effectively illustrate the average behaviour of these gene groups, which is much more easily appreciated in metaplots. Therefore, we have removed the genomic snapshots in Figure 5c and retained the metaplots in Figure 5d (now Figure 5c in revised manuscript) to illustrate this point. Furthermore, we have now more clearly described the very modest distinction between ZC3H4 occupancy at SET1-independent and SET1-dependent genes in the revised text on lines 286-296 to ensure this point is clear to the reader as follows:

'In contrast to SET1 proteins, and consistent with its proposed role in terminating extragenic transcription, ZC3H4 localises to active enhancers that are bound by RNA Pol II but which are not typically associated with CGIs (Fig.5a-b and Supplementary Fig.5b-c). As reported previously, we also found that ZC3H4 localises with RNA Pol II at actively transcribed gene promoters^{78,79,81}, where it is enriched on the shoulders of RNA Pol II peaks corresponding to where sense and antisense early transcription elongation complexes predominate (Fig.5a-b). Therefore, despite previous reports that ZC3H4 primarily affects extragenic and non-coding RNAs, we reasoned that ZC3H4 might also contribute to PTT of protein-coding transcription. Importantly, while ZC3H4 enrichment was similar at both SET1-independent and -dependent genes, it was very slightly biased downstream of TSSs towards the gene body at SET1-dependent genes, suggesting that ZC3H4 might influence SET1-dependent and -independent genes differently (Fig.5c).'

5. Line 179 – “bulk western” should read “bulk western blot”.

We have now edited this in the revised manuscript as suggested.

6. Line 632 – Proteinase K digestion was performed at 45C. Most commonly, this is done at 55C, especially for short incubations such as 1h. Is this a typo, or is 45C correct?

45 C is correct. This Proteinase K digestion temperature (45 C) and time (1 hr) is consistent with a ChIP protocol described by Thomas Milne, Keji Zhao, and Jay Hess¹¹.

Reviewer #3 (Remarks to the Author):

In this study, Hughes and colleagues report a catalytic activity-independent role of the SET1 complex in preventing early termination by WDR82-ZC3H4 at lowly expressed CpG island containing genes. The authors found that loss of SET1A and even more so the combined loss of SET1A/B reduced expression of low-to-moderately transcribed genes in a manner that did not correlate with loss of H3K4me3. Consistent with this observation, they found that artificial tethering of a catalytically inactive SET1 sufficed to increase transcription of a reporter gene and that this activity was dependent on the interaction with WDR82 which was mediated by a short linear motif. Reduced expression of SET1-dependent genes was associated with reduced Pol II occupancy inside genes with unmodified levels at their 5' ends, a finding consistent with increased premature intragenic termination. Critically, these effects were completely counteracted by the depletion of ZC3H4. These data led the authors to propose that while the ZC3H4-WDR82 complex acts unopposed to terminate transcription at

enhancers and at promoter-divergent transcription units, it is efficiently neutralized by the SET1 complex at low-to-moderately expressed CpGi-containing genes, with highly expressed genes being instead constitutively resistant to termination.

Overall, the study provides a conceptually solid model, strongly supported by experimental data, that clarifies the interplay between two central machineries regulating transcription.

There are a few issues, mainly of minor relevance, that the authors may wish to consider to improve their study.

We thank the reviewer for their extremely supportive comments and, in addressing the points raised, we believe the central findings and associated model are now more clearly explained and supported.

1. The data on SET1 tethering to the reporter gene in Fig. 2 and 3 clearly support a catalytic activity-independent effect of SET1 on transcription and they show that, based on the effect of the DPR motif mutant, this effect is WDR82-dependent. However, they do not prove that this effect has anything to do with the prevention of termination, and thus with ZC3H4. It is possible that the effects observed here may merely reflect WDR82-mediated interactions with Pol II rather than the prevention of termination.

We agree with the reviewer that in the context of the tethering experiments we cannot unequivocally arrive at conclusions as to whether the effects we observe are due to the prevention of transcription termination. To ensure this is clear to the reader we have drawn attention to this point in the discussion on lines 393-399 of the revised manuscript as follows:

'Like ZC3H4 complexes, SET1 complexes interact with WDR82, and in reporter gene experiments this interaction appears to be important for the effects of SET1A on gene expression (Fig.3). We cannot rule out that the influence of SET1A/B on reporter gene expression is due to effects beyond counteracting transcription termination. However, based on our genome-wide studies, we propose that the binding of WDR82-containing SET1 complexes can antagonise the activity of ZC3H4/WDR82 complexes, with this being particularly important for the transcription and expression of low to moderately transcribed genes.'

2. The interpretation of the ZC3H4 ChIP-seq data probably exceeds what can be rigorously extracted from the data. As ZC3H4 peaks appear to be very broad, it would be opportune to interpret more cautiously their relationship to the much narrower Pol II peaks and in particular the slight differences observed in the metaplots between SET1-dependent and independent genes (lines 281-290, figure 4A-D).

We agree with the reviewer that the differences in occupancy of ZC3H4 at SET1-independent and SET1-dependent genes is very modest (as also commented on by reviewer 2). Nevertheless, we thought it relevant to draw attention to this as a potential difference in how ZC3H4 behaves at genes that have different transcription levels and sensitivities to SET1 depletion. As suggested by the reviewer we have now drawn attention to the very slight distinction between ZC3H4 occupancy at SET1-independent and SET1-dependent genes on lines 286-296 in the revised text (see below) to ensure this point is clear to the reader. We believe the revised manuscript now deals with these observations in a more cautious manner and we thank the reviewer for suggesting these alterations.

'In contrast to SET1 proteins, and consistent with its proposed role in terminating extragenic transcription, ZC3H4 localises to active enhancers that are bound by RNA Pol II but which are not typically associated with CGIs (Fig.5a-b and Supplementary Fig.5b-c). As reported previously, we

also found that ZC3H4 localises with RNA Pol II at actively transcribed gene promoters ^{78,79,81}, where it is enriched on the shoulders of RNA Pol II peaks corresponding to where sense and antisense early transcription elongation complexes predominate (Fig.5a-b). Therefore, despite previous reports that ZC3H4 primarily affects extragenic and non-coding RNAs, we reasoned that ZC3H4 might also contribute to PTT of protein-coding transcription. Importantly, while ZC3H4 enrichment was similar at both SET1-independent and -dependent genes, it was very slightly biased downstream of TSSs towards the gene body at SET1-dependent genes, suggesting that ZC3H4 might influence SET1-dependent and -independent genes differently (Fig.5c).'

3. The effects of ZC3H4 depletion on protein coding genes were previously reported (ref. 78), with the CpGi-containing ZC3H4 gene being one example of such regulation. The current study, however, has the merit to provide a conceptual framework for such effects. What remains unclear is whether the magnitude of the effects of ZC3H4 depletion observed at genes is comparable to, lower or higher than that observed at extragenic regions.

As suggested by the reviewer we have now examined the magnitude of the effects of ZC3H4 depletion on transcription at genes and extragenic regions. This revealed that the magnitude of the increase in transcription in the absence of ZC3H4 is larger at extragenic regions (upstream antisense transcripts and enhancer RNAs) than at genes. We have now included this new analysis in the revised manuscript in Supplementary Fig.5e (and below in Reviewer Figure 14) and drawn attention to this in the main text of the revised manuscript on lines 306-309 as follows:

'Strikingly, when we analysed genic transcription after ZC3H4 depletion, we observed increased transcription of 2599 genes (Fig.5g-h). This indicates that ZC3H4 also significantly counteracts genic transcription, although these effects were of lesser magnitude than at extragenic regions (Supplementary Fig.5e).'

Reviewer Figure 14

A box plot showing the log2 fold change in transcription (cTT-seq) in the ZC3H4-dTAG line after 2 hours of dTAG13 treatment at genes with increased transcription (UP genes), TSS antisense regions with increased transcription (UP antisense), and enhancers with increased transcription (UP enhancers).

4. While I agree with the main points raised in the discussion, in particular those relative to the unclear regulatory logic of the interplay between SET1 and ZC3H4, I do not agree with the message provided

by the scheme in Figure 6h and in particular with the representation on the genic (right) side. This scheme hints at the idea of a free, non-WDR82-bound ZC3H4 which is somehow prevented from entering in contact with Pol II inside genes, which is not what ChIP-seq data show. One more reasonable model is that the many CTD repeats may accommodate multiple complexes via WDR82-mediated interactions with Ser5P-CTD and that the integration of different signals eventually determine the output.

We thank the reviewer for bringing up this important point about the model figure, particularly with respect to the right-hand side of the model. Our depiction of ZC3H4 not being bound to WDR82 on the genic side of the gene in our model cartoon in Fig.6h of the original submission was an oversight. We had intended to illustrate ZC3H4 associating with WDR82, which we have now remedied in our revised model figure. We also agree with the reviewer that a more likely model is that the CTD can accommodate multiple WDR82-associated complexes and that integration of these opposing signals will ultimately lead to the output. As suggested by the reviewer such a model is consistent with the ZC3H4 ChIP-seq data in the original submission, and new data obtained during revision showing that ZC3H4 occupancy at target sites does not increase after depletion of SET1 complexes (Supplementary Fig.6i). In line with these findings, we have now included a more detailed model figure (as shown below for the reviewer) that alludes to the possibility of both WDR82 complexes engaging with the RNA Pol II CTD and the integration of their distinct activities determining the effect on transcription.

In order for the reader to appreciate this important point, we have revised the text in the figure legend to highlight this possibility as follows:

'The defined mechanism through which SET1 complexes counteract the function of ZC3H4 complexes remains to be determined, but this likely involves both SET1 and ZC3H4 complexes interacting with the CTD of RNA Pol II and the integration of their distinct activities determining the effect on transcription.'

We have also drawn attention to these points in the text of the revised discussion on lines 343-349 as follows:

'These findings are consistent with the idea that SET1 complexes primarily function to antagonise PTT by ZC3H4 complexes. Importantly, this antagonism does not appear to manifest from direct physical competition for binding to RNA Pol II or WDR82, as depletion of SET1 complexes did not lead to increased binding of ZC3H4 at TSSs and WDR82 is in excess to SET1 and ZC3H4 complexes in ESCs (Supplementary Fig.6h and 6i). As such, we envisage that both SET1 and ZC3H4 complexes

interact with the CTD of RNA Pol II and the integration of their distinct activities determines the effect on transcription (Figure 6h).'

We thank the reviewer for highlighting these points. We believe that the description of the revised model will be clear to the reader and that these revisions significantly improve the revised manuscript.

5. The observation that ZC3H4/WDR82 effects are selectively antagonized by SET1 at genes while they are unopposed at extragenic regions is not reported in the abstract. As this represents a major conceptual and mechanistic aspect of this study, it should be properly highlighted.

We agree with the reviewer that this is a very important point and have now drawn attention to this in the final sentence of the abstract of the revised text as follows:

'Unexpectedly, we discover that SET1 complexes enable gene expression by antagonising premature transcription termination by the ZC3H4/WDR82 complex at CpG island-associated genes. In contrast, at extragenic sites of transcription, which typically lack CpG islands and SET1 complex occupancy, we show that the activity of ZC3H4/WDR82 is unopposed. Therefore, we reveal a gene regulatory mechanism whereby CpG islands are bound by a protein complex that specifically protects genic transcripts from premature termination, effectively distinguishing genic from extragenic transcription and enabling normal gene expression.'

1. Ke, S. et al. m(6)A mRNA modifications are deposited in nascent pre-mRNA and are not required for splicing but do specify cytoplasmic turnover. *Genes Dev* **31**, 990-1006 (2017).
2. Blackledge, N.P. et al. PRC1 Catalytic Activity Is Central to Polycomb System Function. *Mol Cell* **77**, 857-874 e9 (2020).
3. Jang, Y., Wang, C., Zhuang, L., Liu, C. & Ge, K. H3K4 Methyltransferase Activity Is Required for MLL4 Protein Stability. *J Mol Biol* **429**, 2046-2054 (2017).
4. Sze, C.C. et al. Coordinated regulation of cellular identity-associated H3K4me3 breadth by the COMPASS family. *Sci Adv* **6**, eaaz4764 (2020).
5. Austenaa, L.M.I. et al. A first exon termination checkpoint preferentially suppresses extragenic transcription. *Nat Struct Mol Biol* **28**, 337-346 (2021).
6. Estell, C., Davidson, L., Steketee, P.C., Monier, A. & West, S. ZC3H4 restricts non-coding transcription in human cells. *Elife* **10**(2021).
7. Brown, D.A. et al. The SET1 Complex Selects Actively Transcribed Target Genes via Multivalent Interaction with CpG Island Chromatin. *Cell Rep* **20**, 2313-2327 (2017).
8. Lee, J.H. & Skalnik, D.G. Wdr82 is a C-terminal domain-binding protein that recruits the Setd1A Histone H3-Lys4 methyltransferase complex to transcription start sites of transcribed human genes. *Mol Cell Biol* **28**, 609-18 (2008).
9. Lee, J.H., You, J., Dobrota, E. & Skalnik, D.G. Identification and characterization of a novel human PP1 phosphatase complex. *J Biol Chem* **285**, 24466-76 (2010).
10. Cao, K. et al. SET1A/COMPASS and shadow enhancers in the regulation of homeotic gene expression. *Genes Dev* **31**, 787-801 (2017).
11. Milne, T.A., Zhao, K. & Hess, J.L. Chromatin immunoprecipitation (ChIP) for analysis of histone modifications and chromatin-associated proteins. *Methods Mol Biol* **538**, 409-23 (2009).

REVIEWERS' COMMENTS

Reviewer #1 (Remarks to the Author):

I have read through the revisions provided by the authors and the comprehensive response to reviewers. While not every experiment could be performed, the authors explained their responses well and I feel that the paper is now ready for publication.

Reviewer #2 (Remarks to the Author):

In my opinion the authors did a great job addressing the concerns raised by the reviewers. Whenever they could, they have performed additional experiments and included the new data in the revised manuscript. At several instances also the textual changes helped to improve the manuscript. The system they have developed to convert endogenous SET1A to a catalytically inactive form is elegant. It is unfortunate that the catalytically dead version appears to be unstable, preventing the authors to perform the experiment requested. Nevertheless, I would strongly urge the authors to clearly mention this as a limitation of their study when discussing their findings. In fact, that inactive SET1A is less abundant than the wild type form is an important finding on its own worth highlighting. With that considered, I can support publication in Nature Communications.

Reviewer #3 (Remarks to the Author):

The authors carefully addressed all remaining issues in their study. The current manuscript provides a compelling description of the regulatory logic controlling the interplay between SET1 complexes and ZC3H4/WDR82 and represents a most valuable addition to the field.